# Statistical inference on representational geometries

**Heiko H Schütt**[1]*[†], **Alexander D Kipnis**[1][‡], **Jörn Diedrichsen**[2],
**Nikolaus Kriegeskorte**[1]*

[1]Zuckerman Institute, Columbia University, New York, United States; [2]Western University, London, Canada

**Abstract** Neuroscience has recently made much progress, expanding the complexity of both neural activity measurements and brain-computational models. However, we lack robust methods for connecting theory and experiment by evaluating our new big models with our new big data. Here, we introduce new inference methods enabling researchers to evaluate and compare models based on the accuracy of their predictions of representational geometries: A good model should accurately predict the distances among the neural population representations (e.g. of a set of stimuli). Our inference methods combine novel 2-factor extensions of crossvalidation (to prevent overfitting to either subjects or conditions from inflating our estimates of model accuracy) and bootstrapping (to enable inferential model comparison with simultaneous generalization to both new subjects and new conditions). We validate the inference methods on data where the ground-truth model is known, by simulating data with deep neural networks and by resampling of calcium-imaging and functional MRI data. Results demonstrate that the methods are valid and conclusions generalize correctly. These data analysis methods are available in an open-source Python toolbox (rsatoolbox.readthedocs.io).

*For correspondence:
hs3110@columbia.edu (HHS);
N.Kriegeskorteg@columbia.edu
(NK)

Present address: [†]Université du Luxembourg, Esch-Belval, Luxembourg; [‡]Max Planck Institute for Biological Cybernetics, Tuebingen, Germany

## Editor's evaluation

Schütt and colleagues introduce a new method for statistical inference on representational geometries based on a cross-validated two-factor bootstrap that allows for generalization across both participants and stimuli while allowing the fitting of flexible models. In a series of elegant simulations and empirical analyses on existing datasets, the authors validate the method statistically. The work provides a fundamental and compelling advance for the analysis of representational geometries.

## Introduction

Experimental neuroscience has recently made rapid progress with technologies for measuring neural population activity. Spatial and temporal resolution, as well as the coverage of measurements across the brains of animals and humans have all improved considerably (*Parvizi and Kastner, 2018*; *Abbott et al., 2020*; *Wang and Xu, 2020*; *Allen et al., 2021*; *Guo et al., 2021*; *Uğurbil, 2021*; *Bandettini et al., 2021*). Activity is measured using a wide range of techniques, including electrode recordings (*Jun et al., 2017*; *Steinmetz et al., 2018*; *Parvizi and Kastner, 2018*), calcium imaging (*Wang and Xu, 2020*), functional magnetic resonance imaging (fMRI; *Allen et al., 2021*; *Uğurbil, 2021*; *Bandettini et al., 2021*), and scalp electro- and magnetoencephalography (EEG and MEG; *Baillet, 2017*; *Craik et al., 2019*). In parallel to the advances in measuring brain activity, theoretical neuroscience has substantially scaled up brain-computational models that implement computational theories (e.g. *Kriegeskorte, 2015*; *Kell et al., 2018*; *Kubilius et al., 2019*; *Zhuang et al., 2021*). The engineering advances associated with deep learning (e.g. *Paszke et al., 2019*; *Abadi et al., 2015*) provide powerful tools for modeling brain information processing for complex, naturalistic tasks (*LeCun et al., 2015*).

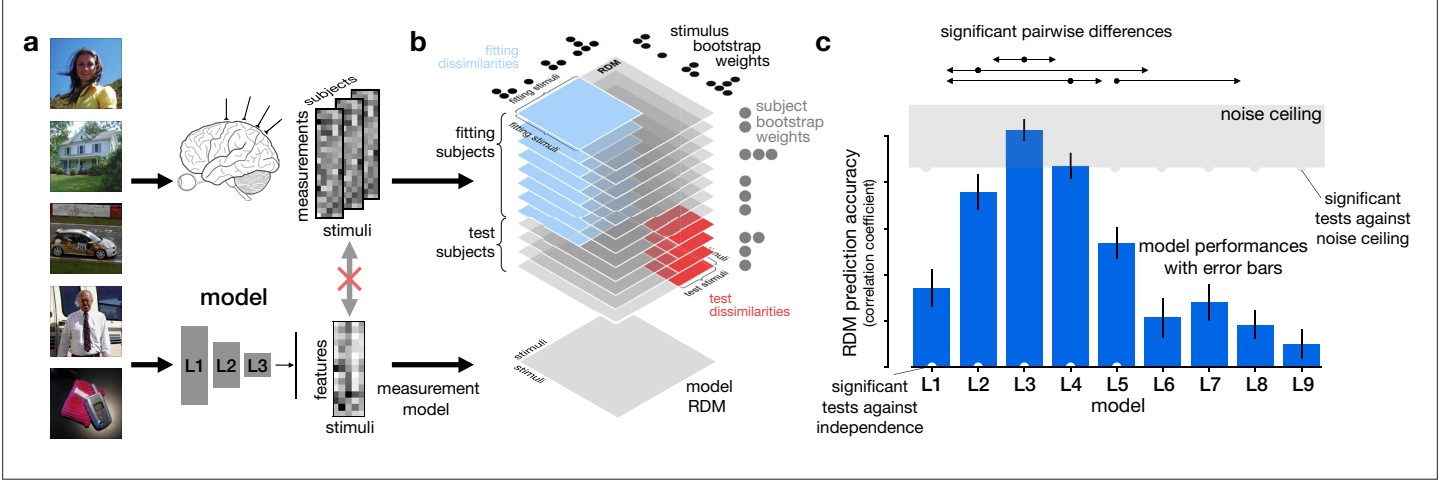

**Figure 1.** Overview of model-comparative inference. (**a**) Multiple conditions are presented to observers and to models (here different stimulus images). The brain measurements during the presentation produce a set of measurements for each stimulus and subject, potentially with repetitions; a model yields a feature vector per stimulus. Importantly, no mapping between brain measurement channels and model features is required. (**b**) To compare the two representations, we compute a representational dissimilarity matrix (RDM) measuring the pairwise dissimilarities between conditions for each subject and each model. For model comparison, we perform 2-factor crossvalidation within a 2-factor bootstrap loop to estimate our uncertainty about the model performances. On each fold of crossvalidation, flexible models are fitted to the representational dissimilarities for a set of fitting stimuli estimated in a set of fitting subjects (blue fitting dissimilarities). The fitted models must then predict the representational dissimilarities among held-out test stimuli for held-out test subjects (red test dissimilarities). The resulting performance estimates are not biased by overfitting to either subjects or stimuli. (**c**) Based on our uncertainty about model performances (error bars indicate estimated standard errors of measurement), we can perform various statistical tests, which are marked in the graphical display. Dew drops (gray) clinging to the lower bound of the noise ceiling mark models performing significantly below the noise ceiling. White dew drops on the horizontal axis mark models whose performance significantly exceeds 0 or chance performance. Pairwise differences are summarized by arrows. Each arrow indicates that the model marked with the dot performed significantly better than the model the arrow points at and all models further away in the direction of the arrow.

Image credit: Ecoset (**Mehrer et al., 2017**) and Wiki Commons.

How to leverage the new big data to evaluate the new big models, however, is an open problem (**Stevenson and Kording, 2011**; **Sejnowski et al., 2014**; **Smith and Nichols, 2018**; **Kriegeskorte and Douglas, 2018**).

An important concept for understanding neural population codes is the concept of *representational geometry* (**Shepard and Chipman, 1970**; **Edelman et al., 1998**; **Edelman, 1998**; **Norman et al., 2006**; **Diedrichsen and Kriegeskorte, 2017**; **Kriegeskorte et al., 2008a**; **Kriegeskorte et al., 2008b**; **Connolly et al., 2012**; **Xue et al., 2010**; **Khaligh-Razavi and Kriegeskorte, 2014**; **Yamins et al., 2014**; **Cichy et al., 2014**; **Haxby et al., 2014**; **Freeman et al., 2018**; **Kietzmann et al., 2019**; **Stringer et al., 2019**; **Chung et al., 2018**; **Chung and Abbott, 2021**; **Kriegeskorte and Wei, 2021**). Neural activity patterns that represent particular pieces of mental content, such as the stimuli presented in a neurophysiological experiment, can be viewed as points in the multivariate neural population response space of a brain region. The representational geometry is the geometry of these points. The geometry is characterized by the matrix of distances among the points. This distance matrix abstracts from the roles of individual neurons and provides a summary characterization of the neural population code that can be directly compared among animals and between brain and model representations (e.g. a cortical area and a layer of a neural network model). The representational geometry provides a multivariate characterization of a neural population code that can be motivated as a generalization of linear decoding analyses. A linear decoder reveals a single projection of the geometry. The full distance matrix (when measured after a transform that renders the noise isotropic) captures what information is available in any linear projection (**Kriegeskorte and Diedrichsen, 2019a**).

A popular method for analyzing representational geometries (**Kriegeskorte and Kievit, 2013**) on which we build here is representational similarity analysis (RSA; **Kriegeskorte et al., 2008a**; **Nili et al., 2014**). RSA is a three-step process (*Figure 1*): In the first step, RSA characterizes the representational geometry of the brain region of interest (ROI) by estimating the representational distance for each pair of experimental conditions (e.g. different stimuli). The distance estimates are assembled

in a representational dissimilarity matrix (RDM). We use the more general term 'dissimilarity' here to include dissimilarity measures that are not distances or metrics in the mathematical sense, such as crossvalidated distance estimators that can return negative values. This relaxation enables inclusion of measures that are not biased by the noise in the data (*Kriegeskorte et al., 2007*; *Nili et al., 2014*; *Walther et al., 2016*; *Kriegeskorte and Diedrichsen, 2019a*), returning values distributed symmetrically about 0, when the true distance is 0, but patterns are noisy estimates. In the second step, each model is evaluated by the accuracy of its prediction of the data RDM. To this end, an RDM is computed for each model representation. Each model's prediction of the data RDM is evaluated using an RDM comparator, such as a correlation coefficient. In the third step, models are inferentially compared to each other in terms of their RDM prediction accuracy to guide computational theory.

RSA is widely used (*Kriegeskorte and Kievit, 2013*; *Haxby et al., 2014*; *Kriegeskorte and Diedrichsen, 2019a*) and has gained additional popularity with the rise of image-computable representational models like deep neural networks (e.g. *Krizhevsky et al., 2012*; *Yamins et al., 2014*; *Khaligh-Razavi and Kriegeskorte, 2014*; *Mehrer et al., 2017*; *Kriegeskorte, 2015*; *Yamins and DiCarlo, 2016*; *Xu and Vaziri-Pashkam, 2021*; *Konkle and Alvarez, 2022*; *Cichy et al., 2016*). There has been important recent progress with methods for estimating representational distances (step 1) as well as measures of RDM prediction accuracy (step 2). For RDM estimation, biased and unbiased distance estimators with improved reliability have been proposed (*Nili et al., 2014*; *Cai et al., 2019*; *Walther et al., 2016*). For quantification of the RDM prediction accuracy, the sampling distribution of distance estimators has been derived and measures of RDM prediction accuracy that take the dependencies between dissimilarity estimates into account have been proposed (*Diedrichsen et al., 2020*). However, existing statistical inference methods for RSA (step 3) have important limitations. Established RSA inference methods (*Nili et al., 2014*) provide a noise ceiling and enable comparisons of fixed models with generalization to new subjects and conditions. However, they cannot handle flexible models, can be severely suboptimal in terms of statistical power, and have not been thoroughly validated using simulated or real data where ground truth is known. Addressing these shortcomings poses three substantial challenges. (1) Model-comparative inference with generalization to new conditions is not trivial because new conditions extend an RDM and the evaluation depends on pairwise dissimilarities, thus violating independence assumptions. (2) Standard methods for statistical inference do not handle multiple random factors — subjects and conditions in RSA. (3) Flexible models, that is models that have parameters enabling them to predict different RDMs, are essential for RSA (*Diedrichsen et al., 2018*; *Kriegeskorte and Diedrichsen, 2016*). Evaluation of such models requires methods that are unaffected by overfitting to either subjects or conditions to avoid a bias in favor of more flexible models.

Here, we introduce a comprehensive methodology for statistical inference on models that predict representational geometries (*Figure 1*). We introduce novel bootstrapping methods that support generalization of model-comparative statistical inferences to new subjects, new conditions, or both simultaneously, as required to support the theoretical claims researchers wish to make. We also introduce a novel crossvalidation method for estimation of the RDM prediction accuracy of flexible models, that is models with parameters fitted to the data (*Khaligh-Razavi and Kriegeskorte, 2014*; *Kriegeskorte and Diedrichsen, 2016*). This is important, because theories do not always make a specific prediction for the representational geometry. There may be unknown parameters, such as the relative prevalences of different tuning functions (*Khaligh-Razavi and Kriegeskorte, 2014*; *Jozwik et al., 2016*) in the neural population or properties of the measurement process (*Kriegeskorte and Diedrichsen, 2016*). The combination of our 2-factor bootstrap and 2-factor crossvalidation methods enables statistical comparisons among fixed and flexible models that generalize across subjects and conditions.

We thoroughly validate the new inference methods using simulations and neural activity data. Extensive simulations based on deep neural network models and models of the measurement process enable us to test model-comparative inference in a setting where the ground-truth model (the one that actually generated the data) is known. These simulations confirm the validity of the inference procedures and their ability to generalize to the populations of subjects and/or conditions. We also validated the methods on real data from calcium imaging (mouse) and functional MRI (human). For both datasets, we confirm that conclusions generalize from an experimental dataset (a subset of the real data) to the entire dataset (which serves as a stand-in for the population). The statistical inference

methodology described in this paper is available in a new open-source RSA toolbox written in Python (https://github.com/rsagroup/rsatoolbox, copy archived at *Schütt, 2023*).

## Results

We now introduce the 2-factor bootstrap procedure for model-comparative inference and the 2-factor crossvalidation procedure for unbiased evaluation of flexible models. This paper also introduces a new representational dissimilarity estimator for electrophysiological recordings of patterns of firing rates across a population of neurons, based on the KL-divergence between Poisson distributions (Appendix 2) and a faster alternative to the rank correlation $\tau_a$ as an RDM comparator (*Nili et al., 2014*), which we call $\rho_a$ (Appendix 3). The proposed inferential methods work for any representational dissimilarity measure and any RDM comparator. We evaluate alternative RDM comparators in terms of their power in Appendix 6. A complete description of all steps of the new methodology can be found in the Materials and methods (Full description of the RSA method).

### Methods for inference on representational geometries

A simple approach to inferential comparison of two models is to compute the difference between the models' performance estimates for each subject and use Student's $t$-test (or a nonparametric alternative). However, inference then only takes the variability over subjects into account and thus does not justify generalization to different experimental conditions (e.g. different stimuli). Computational neuroscience usually pursues insights that generalize not only to a population of subjects but also to a population of conditions (*Yarkoni, 2020*). To support generalization to the population of conditions statistically, we require uncertainty estimates that treat the experimental conditions as a random sample from a population (*Kriegeskorte et al., 2008a*), whether or not the subjects are treated as a random sample.

For frequentist inference, the challenge is to estimate how variable the model-performance estimates would be if we repeated the experiment many times with new subjects and/or conditions. We would like to know (1) the variance of each model's performance estimate and (2) the variance of the estimated performance difference for each pair of models. The variance of model-performance estimates enables us to statistically compare each model to a fixed value such as an RDM correlation of 0. The variance of our estimate of model-performance difference enables us to statistically compare two models to each other (see Frequentist tests for model evaluation and model comparison for details).

### Estimating the variance of model-performance estimates for generalization to new subjects and conditions

To estimate the variance of model-performance estimates across repetitions of the experiment with new conditions, we use a bootstrap method. Bootstrap methods estimate the variance of experimental outcomes by sampling from the measured data with replacement, treating the measured data as an approximation to the population (*Efron and Tibshirani, 1994*). The population here is the set of experimental conditions of which the actual experimental conditions can be considered a random sample. Because the conditions do not have independent influences on the model evaluations, we cannot compute a sample variance across conditions as we can across subjects to replace the bootstrap.

When we bootstrap-resample conditions, we obtain RDMs of the same size as the original RDMs, but some of the conditions will be repeated. Here, we exclude the entries that correspond to the dissimilarity of any condition with itself from the comparisons between RDMs. Simulations confirm that this procedure yields a good estimate of how variable the results are when we sample new conditions with the same subjects (Figures 4a and 6g).

For simultaneous generalization to the populations of both conditions and subjects, we can employ a 2-factor bootstrap (*Figure 1b*) as introduced previously (*Nili et al., 2014*; *Storrs et al., 2021*). However, our simulations and theory here show that a naive 2-factor bootstrap approach triple-counts the variance contributed by the measurement noise (Methods, Estimating the uncertainty of our model-performance estimates, Figures 4c and 7c). This effect is not unique to RSA; a naive 2-factor bootstrap will triple-count variance related to the measurement noise for any type of experiment in which two factors (here subject and condition) jointly determine the experimental outcome. The

true variance $\sigma_{both}^2$ of the experimental outcome when sampling both factors can be separated into a contribution from condition sampling ($\sigma_{cond}^2$), a contribution from subject sampling ($\sigma_{subj}^2$), and a contribution of the interaction of subjects and conditions or measurement noise ($\sigma_{noise}^2$).

$$\sigma_{both}^2 \approx \sigma_{subj}^2 + \sigma_{cond}^2 + \sigma_{noise}^2 \tag{1}$$

This decomposition is for the actual variance $\sigma_{both}^2$ across repeated experiments with new subjects and conditions. The variance $\hat{\sigma}_{both}^2$ of the naive 2-factor bootstrap can likewise be decomposed into three additive terms (Online Methods, Estimating the uncertainty of our model-performance estimates), corresponding to subject sampling, condition sampling, and the interaction and/or noise. However, in the naive 2-factor bootstrap estimate $\hat{\sigma}_{both}^2$, the independent noise contribution enters not only its own term, but also the two others. Thus, the original bootstrap estimate contains the noise variance component three times instead of once:

$$\hat{\sigma}_{both}^2 \approx (\sigma_{subj}^2 + \sigma_{noise}^2) + (\sigma_{cond}^2 + \sigma_{noise}^2) + \sigma_{noise}^2$$
$$= \sigma_{subj}^2 + \sigma_{cond}^2 + 3\sigma_{noise}^2 \tag{2}$$

This problem can be understood by considering the 1-factor bootstraps, which also contain the independent noise component although it has not been added explicitly:

$$\hat{\sigma}_{subj}^2 \approx \sigma_{subj}^2 + \sigma_{noise}^2 \tag{3}$$

$$\hat{\sigma}_{cond}^2 \approx \sigma_{cond}^2 + \sigma_{noise}^2 \tag{4}$$

When we bootstrap two factors, this automatic inclusion of the noise component happens three times. We confirmed this by both theory and simulation. The overestimate of the variance renders the naive 2-factor bootstrap conservative and not optimally powerful.

To correct the variance estimate, we introduce a novel corrected 2-factor bootstrap procedure to estimate the variance: We first compute the 1-factor bootstrap variance estimates $\hat{\sigma}_{subj}^2$ and $\hat{\sigma}_{cond}^2$. We also compute the naive 2-factor bootstrap estimate $\hat{\sigma}_{both}^2$. We can then linearly combine the variances from these three bootstraps to cancel the surplus contribution from the measurement noise. This procedures yields a corrected 2-factor bootstrap estimate $\hat{\sigma}_{c2f}^2$ that has approximately the right expected value:

$$\hat{\sigma}_{c2f}^2 = 2(\hat{\sigma}_{subj}^2 + \hat{\sigma}_{cond}^2) - \hat{\sigma}_{both}^2$$
$$\approx \sigma_{subj}^2 + \sigma_{cond}^2 + \sigma_{noise}^2 \tag{5}$$

The approximations in these equations are due to $\frac{N-1}{N}$ factors that apply to the individual terms. We give the exact formulae including these factors in the methods section (Estimating the uncertainty of our model-performance estimates). We show in multiple simulations that this estimate approximates the correct variance better than the uncorrected 2-factor bootstrap (Figures 4c and 7c).

To stabilize the estimator and eliminate the possibility of a negative variance estimate, we bound the estimate from above and below. We use both $\hat{\sigma}_{subj}^2$ and $\hat{\sigma}_{stim}^2$ as lower bounds for the estimate as the variances they estimate are always smaller than the true variance. As an upper bound, we use $\hat{\sigma}_{both}^2$, the naive, conservative estimate. Bounding slightly biases the variance estimate, but reduces its variability and ensures that it is strictly positive.

## Evaluating the performance of flexible models

We often want to test *flexible models*, that is models that have parameters to be fitted to the brain-activity data. Two elements that often require fitting are weights for the model features and parameters of a measurement model. Feature weighting is required when a model is not meant to specify a priori how prevalent different tuning profiles are in the neural population or in the measured signals. For example, for deep neural network representations to match brain responses well, it is usually necessary to weight the features (e.g. *Yamins et al., 2014*; *Khaligh-Razavi and Kriegeskorte, 2014*; *Khaligh-Razavi et al., 2017*; *Storrs et al., 2021*). A flexible measurement model may be necessary to

account for the process of measurement, which may subsample, average, or distort neural responses. For example, fMRI voxels average the neural activity locally, which can be modeled with a parameter for the local averaging range, and electrophysiological recordings may preferentially sample certain classes of neurons (*Kriegeskorte and Diedrichsen, 2016*).

To avoid the bias in the model-performance estimates that can result from overfitting of flexible models, we use crossvalidation. Crossvalidation means that we partition the dataset into separate test sets. In each fold of crossvalidation, we then fit the models to all but one set and evaluate on the held-out set. Taking the average over the folds yields a single performance estimate. As for bootstrapping, crossvalidation is performed over both conditions and subjects so as to avoid overestimating the generalization performance of flexible models when tested on new subjects and new conditions drawn from the populations of subjects and conditions sampled in the actual experiment (*Figure 1b*).

Because the RDM for the test set must contain multiple values to allow a sensible comparison, the smallest possible number of conditions to perform crossvalidation is 6, which would yield three test conditions for twofold crossvalidation. For small numbers of conditions, we use twofolds. We use threefolds for ≥12 conditions, fourfolds for ≥24 conditions, and fivefolds for ≥40 conditions. These numbers seem to work reasonably well, but were chosen ad hoc.

To estimate our uncertainty about the crossvalidated model performances, we use the same bootstrap methods as for fixed models. To do so, we need to perform crossvalidation on each bootstrap sample. We call this procedure bootstrap-wrapped crossvalidation.

In any crossvalidation, different ways to partition the data into test sets lead to different overall evaluations of the models. When we partition the conditions set into disjoint test sets in RSA, this effect is particularly strong, because dissimilarities between conditions in separate test sets do not contribute to the evaluation in any fold. The variance in the evaluations created by this random assignment is generated by our analysis and would vanish if we performed repeated cycles of crossvalidation with all possible partitionings of the conditions set into test sets. Unfortunately, such exhaustive crossvalidation will usually be prohibitively expensive in terms of computation time, especially in bootstrap-wrapped crossvalidation.

We can estimate the variance without this surplus by sampling $n_{cv} > 1$ different randomly chosen partitionings of the conditions set into crossvalidation test sets for each bootstrap sample. Each of the $n_{cv}$ partitionings into $k$ subsets defines a complete cycle of $k$-fold crossvalidation. The bootstrap-wrapped crossvalidation estimate of the variance of the model-performance estimates with $n_{cv}$ crossvalidation cycles will be larger than the variance $\sigma_{boot}^2$ of the exact mean performance over all possible partitionings of a dataset. When we assume that the variance $\sigma_{cv}^2$ of randomly chosen partitionings around the mean is equal for each bootstrap sample, the overall variance $\sigma_{bootcv,n_{cv}}^2$ is:

$$\sigma_{bootcv,n_{cv}}^2 = \sigma_{boot}^2 + \frac{\sigma_{cv}^2}{n_{cv}} \tag{6}$$

When we have more than one cycle of crossvalidation for each bootstrap sample, it is straightforward to compute an estimate for the variance we would have gotten if we had drawn only a single partitioning $\sigma_{bootcv,1}^2$. We can simply use only the $i$ th partitioning for each bootstrap to estimate the variance and average these estimates. Using these two variance estimates for 1 and $n_cv$ partitionings, we can simply solve for the variance contributions of the random partitioning and of the bootstrap:

$$\hat{\sigma}_{cv}^2 = \frac{n_{cv}}{n_{cv} - 1} \left( \hat{\sigma}_{bootcv,1}^2 - \hat{\sigma}_{bootcv,n_{cv}}^2 \right) \tag{7}$$

$$\hat{\sigma}_{boot}^2 = \hat{\sigma}_{bootcv,n_{cv}}^2 - \frac{\hat{\sigma}_{cv}^2}{n_{cv}}$$

$$= \hat{\sigma}_{bootcv,n_{cv}}^2 - \frac{\hat{\sigma}_{bootcv,1}^2 - \hat{\sigma}_{bootcv,n_{cv}}^2}{n_{cv} - 1} \tag{8}$$

Thus, we can directly compute an estimate of the variance we expect for exhaustive crossvalidation from two or more crossvalidation cycles using random partitionings for each bootstrap sample. The repetition across bootstrap samples enables a stable estimate even for $n_{cv} = 2$. The average estimate is independent of $n_{cv}$ (*Figure 2a*). We could invest computation in increasing either the number of bootstrap samples or the number of crossvalidation cycles per bootstrap sample. Our simulations

show that the reliability of the bootstrap estimate of the variance of the model-performance estimate improves more when we increase the number of bootstrap samples than when we increase the number of crossvalidation cycles per bootstrap sample (*Figure 2b*). Thus, we recommend using only two crossvalidation cycles per bootstrap sample.

This crossvalidation approach provides model-performance estimates that are not biased by overfitting of flexible models to either subjects or conditions. Fixed and flexible models with different numbers of parameters can be robustly compared with generalization over conditions and/or subjects. The method can handle any model that can be fitted efficiently enough (for the types of flexible models we actually implemented, see Methods, Flexible models).

## Validation of the statistical inference methods

We validate the inference methods using simulations, functional MRI data, and neural data. First, we establish that the statistical tests for model comparison are valid, controlling the false-positive rate at the nominal level. This requires simulating data under the null hypothesis, where two models that predict distinct RDMs are exactly equal in their RDM prediction accuracy. We use a matrix-normal model to simulate this null scenario for model comparison. Second, we show that the estimates of our uncertainty about model performance correctly capture the true variability for different generalization schemes in more realistic simulated scenarios based on neural network models. In these simulations, we cannot simulate the null hypothesis of two models that predict the representational geometry equally accurately. We also use these more realistic simulations to evaluate the power afforded by different RDM comparators. Third, we validate the inference procedure for flexible models, confirming that our bootstrap-wrapped crossvalidation scheme correctly accounts for the overfitting of flexible models. Fourth, we validate the methods using real data, acquired with functional MRI in humans and calcium imaging in mice.

### Validity of inferential model comparisons

A frequentist test is valid when the rate of false positives (i.e. the rate of positive results when the null hypothesis is true) does not exceed the specified error rate $\alpha$ (e.g. 5%). Here, we check the validity of model-comparative inference, where the null hypothesis is that the two models perform equally well at explaining the representational geometry. We simulate scenarios where two models predict distinct geometries, but perform equally well on average at predicting the true representational geometry.

To simulate situations where two different models perform equally well, we generated condition-response matrices (containing an activity level for each combination of condition and response channel) by sampling from matrix-normal density models. A matrix-normal distribution over matrices yields matrices with normally distributed cells whose covariance is separable into a covariance matrix across rows and one across columns. In our case, rows correspond to the experimental conditions (e.g. stimuli) and the columns correspond to measurement channels (e.g. neurons or voxels). For matrix-normal data, the covariance across conditions captures the similarity among condition-related response patterns and determines the expected squared Euclidean-distance RDM (***Diedrichsen and Kriegeskorte, 2017***). The covariance among channels only scales the covariance of the distance estimates. This relationship enables us to generate matrix-normal data for arbitrary choices of the expected squared Euclidean-distance RDM. To model the null hypothesis, we choose two models that predict distinct RDMs and generate data, such that the expected data RDM has equal Pearson correlation to both model RDMs (results in *Appendix 1—figure 1*; details in Appendix 1).

We first evaluated the bootstrap in the scenario, where the goal is to generalize across subjects only. All model-comparative subject-only bootstrap tests were found to be valid (*Appendix 1—figure 1*). Inflated false-positive rates were observed for subject-only bootstrap tests only when using a small sample of subjects (<20). For a small number of samples, bootstrapping is known to produce underestimates of the variance by a factor $\frac{n}{n-1}$ for $n$ samples (e.g. ***Efron and Tibshirani, 1994***, chapter 5.3). In this scenario, we recommend using a $t$-test across subjects, which is more computationally efficient and more accurate than bootstrap methods for small numbers of subjects.

Next, we tested bootstrapping for generalization to new conditions. In this scenario, the bootstrap methods were all conservative, showing false-positive rates substantially below 5% (*Appendix 1—figure 1*). This is expected, because we did not include any random selection of conditions in our data simulation, but enforced the $H_0$ exactly for the measured conditions.

To assess how problematic it is to choose an inference method that ignores the variance due to condition sampling, we ran a simulation in which we sampled the conditions from a large pool. We generated two models that perform equally well on 1000 conditions using matrix-normal sampling and then sampled a smaller set of these conditions for the simulated experiment. In these simulations, all techniques that only take subjects into account as a random factor fail catastrophically (*Appendix 1—figure 1*), with false-positive rates growing with the number of simulated subjects and reaching 60% at 40 simulated subjects. In contrast, our bootstrap tests that include condition sampling all remain valid, including the uncorrected 2-factor bootstrap and our new corrected 2-factor bootstrap with false-positive rates below the nominal 5%. However, the uncorrected 2-factor bootstrap was extremely conservative.

We also validated the tests against chance performance, where a single model is tested and the null hypothesis is that its performance is at chance level. To do so, we performed similar matrix-normal data simulations, evaluating a model that predicts a specific randomly sampled RDM on matrix-normal data consistent with an independently sampled random expected data RDM. Results show that a *t*-test across subjects as well as the bootstrap *t*-test approaches provide valid inference (*Appendix 1—figure 1*, top row). The subject *t*-test and the corrected 2-factor bootstrap *t*-test avoid overly conservative false-positive rates.

We conclude that the tests are valid in these simple simulated $H_0$ scenarios, where we are able to estimate the false-positive rate. In more realistic simulations using neural network models and real data, we can no longer simulate distinct models that predict the data RDM equally well. We therefore restrict ourselves to evaluating our bootstrap estimate of the variance of model-performance estimates, assuming that the false-positive rates are adequately controlled when we use an accurate variance estimate.

## Criteria for evaluation of inference procedures

To evaluate alternative inference procedures, we perform simulations that reveal (1) whether the estimates of the uncertainty of the model-performance estimates are accurate (ensuring the validity of the inferences), and (2) how sensitive different model comparison methods are to subtle differences between models (determining the power of the inferences). To measure whether our bootstrap methods correctly estimate the uncertainty of the model-performance estimates, we compute the relative uncertainty (RU). The RU is the standard deviation of the bootstrap distribution of

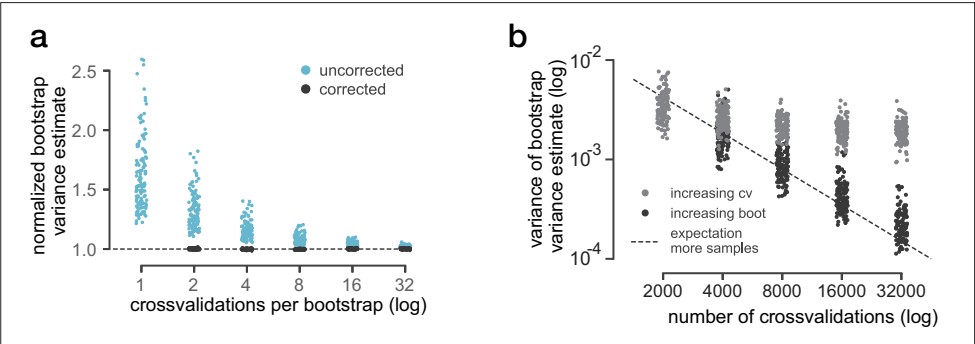

**Figure 2.** Correction for variance caused by crossvalidation. (**a**) Unbiased estimates of the variance of model-performance estimates (dashed line) require either many crossvalidation cycles (light blue dots) or the proposed correction formula (back dots). Each model in each simulated dataset contributes one dot to each point cloud in this plot, corresponding to the average estimated variance across 100 repeated analyses. All variance estimates of a model are divisively normalized by the average corrected variance estimate for this model over all numbers of crossvalidation cycles for the dataset. For many crossvalidation cycles, the uncorrected and corrected estimates converge, but the correction formula yields this value even when we use only two crossvalidation cycles. (**b**) Reliability of the corrected bootstrap variance estimate across multiple estimations on the same dataset, comparing the use of more crossvalidation cycles per bootstrap sample (gray, 2, 4, 8, 16, 32 crossvalidations at 1000 bootstrap samples) to using more bootstrap samples (black, 1000, 2000, 4000, 8000, 16,000 bootstrap samples with 2 crossvalidation cycles per sample). The horizontal axis represents the total number of

*Figure 2 continued on next page*

*Figure 2 continued*

crossvalidation cycles (number of cycles per bootstrap × number of bootstraps). More bootstrap samples are more efficient at stabilizing our bootstrap estimates of the variance of model-performance estimates. Increasing the number of bootstraps decreases the variance roughly at the $N^{-\frac{1}{2}}$ rate expected for sampling approximations indicated by the dashed line.

model-performance estimates $\sigma_{boot}$ divided by the true standard deviation of model-performance estimates $\sigma_{true}$ as observed over repeated simulations:

$$\text{RU} = \frac{\sigma_{boot}}{\sigma_{true}} = \sqrt{\frac{\frac{1}{N}\sum_{i=1}^{N}\sigma_i^2}{\sigma_{true}^2}}, \tag{9}$$

where $\sigma_i^2$ is the variance estimator of the bootstrap in simulated dataset $i$ of the $N$ simulations. Ideally, we would like the bootstrap-estimated variance to match the true variance such that the RU is 1.

To measure how sensitive our analysis is to differences in model performance (e.g. comparing layers of a deep neural network), we define the model discriminability as a signal-to-noise ratio (SNR). The signal is the magnitude of model-performance differences, which is measured as the variance across models of their average of performance estimates across simulations. The noise is the nuisance variation, which includes subject and condition sample variation along with measurement noise. The noise is measured as the average across models of the variance of performance estimates across simulations. This results in the following formula, in which $\text{Perf}_{i,m}$ is the performance of model $m$ of $M$ in repetition $i$ of $N$ repetitions of the simulation:

$$\text{SNR} = \frac{\text{Var}_m\left(\frac{1}{N}\sum_{i=1}^{N}\text{Perf}_{i,m}\right)}{\frac{1}{M}\sum_{i=1}^{M}\text{Var}_i(\text{Perf}_{i,m})}. \tag{10}$$

A higher SNR indicates greater sensitivity to differences in model performance: differences between models are larger relative to the variation of model-performance estimates over repeated simulations.

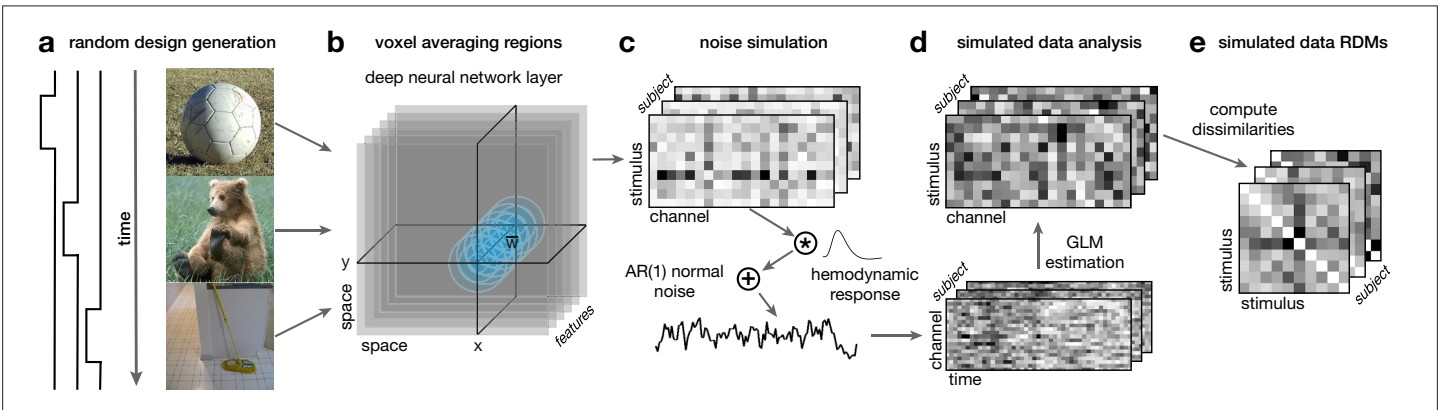

**Figure 3.** Illustration of the deep-neural-network-based simulations for functional magnetic resonance imaging (fMRI)-like data. The aim of the analyses was always to infer which layer of AlexNet the simulation was based on. (**a**) Stimuli are chosen randomly from ecoset (***Mehrer et al., 2017***) and we simulate a simple rapid event-related experimental design. (**b**) 'True' average response per voxel to a stimulus are based on local averages of the internal representations of AlexNet. To simulate the response of a voxel to a stimulus we choose a (x,y)-position uniformly randomly and take a weighted average of the activities around that location. As weights we choose a Gaussian in space and independently draw a weight per feature between 0 and 1. (**c**) To generate a simulated voxel timecourse we generate the undistorted timecourses of voxel activities, convolve them with a standard hemodynamic-response function and add temporally correlated normal noise. (**d**) To estimate the response of a voxel to a stimulus we estimate a standard general linear model (GLM) to arrive at a noisy estimate of the true channel responses we started with in C. (**e**) From the estimated channel responses we compute the stimulus by stimulus dissimilarity matrices. These dissimilarity matrices can then be compared to the dissimilarity matrices computed based on the full deep neural network representations from the different layers.

Note that this measure does not depend on the accuracy of the bootstrap because the bootstrap estimates of the variances do not enter this statistic. The SNR exclusively measures how large differences between models are compared to the level of nuisance variation we simulate, which may include random sampling of conditions, subjects, or both (in addition to measurement noise).

## Validity of generalization to new subjects and conditions

To test whether our inference methods correctly generalize to new subjects and conditions, we performed a simulation that includes random sampling of both subjects and conditions (**Figure 3**). We used the internal representations of the deep convolutional neural network model AlexNet (**Krizhevsky et al., 2012**) to generate fMRI-like simulated data. In each simulated scenario, one of the layers of AlexNet served as the true (data-generating) model, while all layers were considered as candidate models in the inferential model comparisons. We simulated true voxel responses as local averages of the activities of close-by units in the feature maps of layers of the model. The response of each simulated voxel was a local average of unit responses, weighted according to a 2D Gaussian kernel over the locations of the feature map multiplied by a vector of nonnegative random weights (drawn uniformly from the unit interval) across the features. We then simulated hemodynamic-response timecourses and added measurement noise. The covariance structure of the noise was determined by the overlap of the simulated voxels' averaging regions over space and a first-order autoregressive model over time. The simulated data were subjected to a standard general linear model (GLM) analysis to estimate the condition-response matrix. Variation over conditions was generated by using randomly sampled natural images from ecoset (**Mehrer et al., 2017**) as input to the AlexNet model. Variation over subjects was generated by randomly choosing a new location and a new vector of feature weights for each voxel of a new simulated subject.

We simulated N = 100 datasets for each parameter setting to estimate how variable the model-performance estimates truly are. In analysis, we must estimate our uncertainty about model performance from a single dataset. To estimate how accurate these estimates were, we compared the uncertainty estimates used by different inference procedures (including different bootstrap methods)

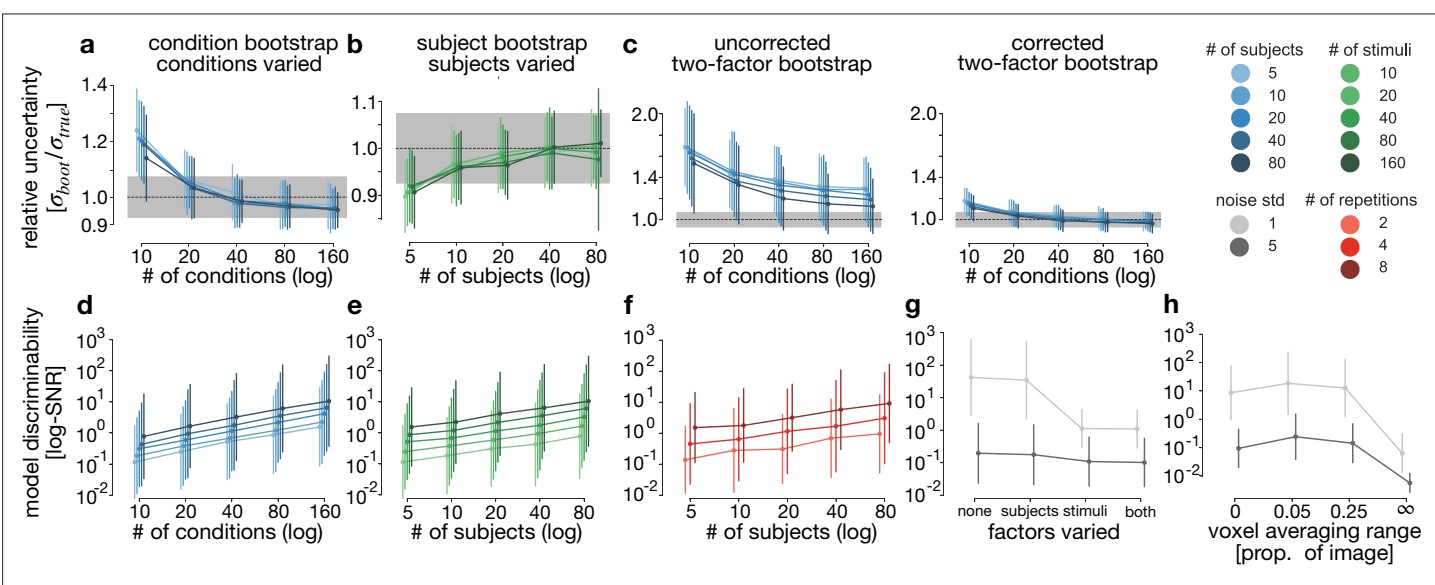

**Figure 4.** Results of the deep-neural-network-based simulations. (**a–c**) Relative uncertainty, that is the bootstrap estimate of the standard deviation of model-performance estimates divided by the true standard deviation over repeated simulations. Dashed line and gray box indicate the expected value and standard deviation due to the number of simulations per condition. (**a**) Bootstrap resampling of conditions when repeated simulations use random samples of conditions and a fixed set of subjects. (**b**) Bootstrap resampling of subjects when simulations use random samples of subjects (simulated voxel placements) and a fixed set of conditions. (**c**) Direct comparison of the uncorrected and corrected 2-factor bootstraps (see Estimating the uncertainty of our model-performance estimates for details) for simulations that varied both conditions and subjects. (**d–h**) Signal-to-noise ratio (**Equation 10**), a measure of sensitivity to differences in model performance, for the different inference procedures and simulated scenarios. Infinite voxel averaging range refers to voxels averaging across the whole feature map. All error bars indicate standard deviations across simulation types that fall into the category.

to the true variability. This comparison is a enables us to validate our inference despite the fact that we cannot compute false-positive rates of the model comparison tests. Our neural-network-based simulations do not contain situations that correspond to the $H_0$ of two different models with equal performance, which would require that the data-generating neural network layer predicts an RDM equally similar to those predicted by two other model layers. As expected, the rate of erroneously finding an alternative model outperforming the true data-generating model was very low (not shown) whenever the type of bootstrap matches the simulated level of generalization because the true layer has a higher average performance than the other models. At the 5% uncorrected significance level, the proportion of cases where any other layer performed significantly better than the true (data-generating) layer was only 1.524%. This rate reflects the differences between the layers of AlexNet, the simulated variability due to subject, stimulus, and voxel sampling, the simulated noise level, and the number of layers. Tests against the best other layer (chosen based on all data) significantly favor this other layer in only 0.694% of cases. Multiple-comparison correction would reduce these model-selection error rates even further.

To test generalization to either new conditions or new subjects (but not both simultaneously), we kept the other dimension constant. When simulating condition sampling, the true variance across conditions is accurately estimated for 40 or more conditions (*Figure 4a*) and is overestimated by 1-factor bootstrap resampling of conditions (rendering the inference conservative) when we have less than about 40 conditions (*Figure 4a*). When simulating subject sampling, the true variance across subjects is accurately estimated for 20 or more subjects (*Figure 4b*) and is underestimated by 1-factor bootstrap resampling of subjects (invalidating the inference) when we have very few subjects (*Figure 4b*). This downward bias corresponds to the $\frac{n}{n-1}$ factor between the sample variance and the unbiased estimate for the population variance. Our implementation in the RSA toolbox uses this factor to correct the variance estimate.

To test our corrected 2-factor bootstrap method's ability to generalize to new subjects and new conditions simultaneously, we simulated sampling of both conditions (stimuli) and subjects in our simulations. The corrected 2-factor bootstrap estimates the overall variation caused by random sampling of subjects and conditions and by measurement noise much more accurately than the naive 2-factor bootstrap (*Figure 4c*). Cases where an incorrect model (not the data-generating model) significantly outperformed the true model occurred in only 0.3% of simulations with the corrected 2-factor bootstrap, even without any multiple-comparison correction. This proportion would be larger if the alternative models performed more similar to the true model than simulated here. The RSA toolbox adjusts for multiple comparisons, controlling either the familywise error rate or the false-discovery rate across all pairwise model comparisons.

Overall, we found that the new more powerful corrected 2-factor bootstrap method yields accurate estimates of the variance across the simulated populations of subjects and conditions when the dataset is large enough (≥20 subjects, ≥40 conditions) and the type of bootstrap matches the population sampling simulated (subject, condition, or both).

The model discriminability (SNR) increases monotonically with the number of measurements, affording greater power for model-comparative inference. Model discriminability increases with the amount of data according to a power law (straight line in log–log plot; *Figure 4d–f*). Such a relationship holds whether we increase the number of conditions, the number of subjects, or the number of repetitions per condition. This result is expected and validates the SNR as an indicator of model-comparison power. In general, increasing the number of measurements helps most for the factor that causes most variability of the performance estimates, rendering generalization harder. For example, in our deep-neural-network-based simulations, the variability over subjects is smaller than the variability across conditions (*Figure 4g*). In this simulation, it thus increases statistical power more to collect data for more conditions. When there is more variability across subjects, the opposite is expected to hold. An intermediate voxel size (Gaussian kernel width) yielded the highest model-performance discriminability as measured by the SNR (*Figure 4h*, see Appendix 5 for more discussion on this topic).

## Validity of inference on flexible models

To validate inferential model comparisons involving flexible models, we made a variant of the deep neural network simulation in which we do not assume to know how voxels average local neural responses. As the simulated ground truth, we set the spatial weights for each voxel to a Gaussian

with a standard deviation of 5% of the image size (full width at half maximum FWHM ≈ 11.77%) and randomly weighted the feature maps (with weights drawn independently for each voxel and feature map uniformly at random from the unit interval; details in Methods, Neural-network-based simulation).

We then used models that a researcher could generate without knowing the ground truth of how voxels average local features. As building blocks for the models, we computed RDMs for different voxel averaging pool sizes and for different methods to deal with averaging across feature maps. To capture voxel averaging across retinotopic locations, we smoothed the feature maps with Gaussians of different sizes. To capture voxel averaging across feature maps, we (1) generated RDMs computed after taking the average across feature maps at each location (avg), (2) computed the expected RDM for the weight sampling implemented in the simulation (weighted), or (3) computed RDMs without any feature-map averaging (full).

We combined these building blocks into two types of flexible model: *selection models* and *nonnegative linear-combination models*. In a selection model, fitting is implemented as selection of the best among a finite set of RDMs. Here we defined one selection model for each method of combining the feature maps. Each selection model contained RDMs computed for different sizes of the local averaging pool. In linear-combination models, fitting consists in finding nonnegative weights for a set of basis RDMs, so as to maximize RDM prediction accuracy. The RDMs contain estimates of the squared Mahalanobis distances, which sum across sets of tuned neurons that jointly form a population code. As component RDMs, we chose the four extreme cases of RDM generation: no pooling across space or averaging across the whole image, each paired with either 'full' or 'avg' treatment of the feature maps. The resulting four-RDM-component linear model approximates the effect of computing the RDM from voxels that reflect the average activity over retinotopic patches of different sizes (*Kriegeskorte and Diedrichsen, 2016*). For the averaging across feature maps, which uses random weights, there is a strong motivation for using a linear model: When the voxel activities are nonnegatively weighted averages of the underlying neurons with the weights drawn independently from the same distribution, the expected squared Euclidean RDM is exactly a linear combination of the RDM computed based on the univariate population-average responses and the RDM based on all neurons (Appendix 4; see also *Carlin and Kriegeskorte, 2017*). For comparison, we also included fixed RDM models, corresponding to component RDMs of the fitted models.

We found that our bootstrap-wrapped crossvalidation (corrected 2-factor bootstrap with adjustment for excess crossvaldation variance) yielded accurate estimates of the uncertainty. The relative uncertainties were close to 1 (*Figure 5b*). The model-performance discriminability (SNR) was primarily determined by how accurately the different models were able to recreate the true measurement model (*Figure 5c*). The highest SNRs were achieved when the assumed model matched exactly

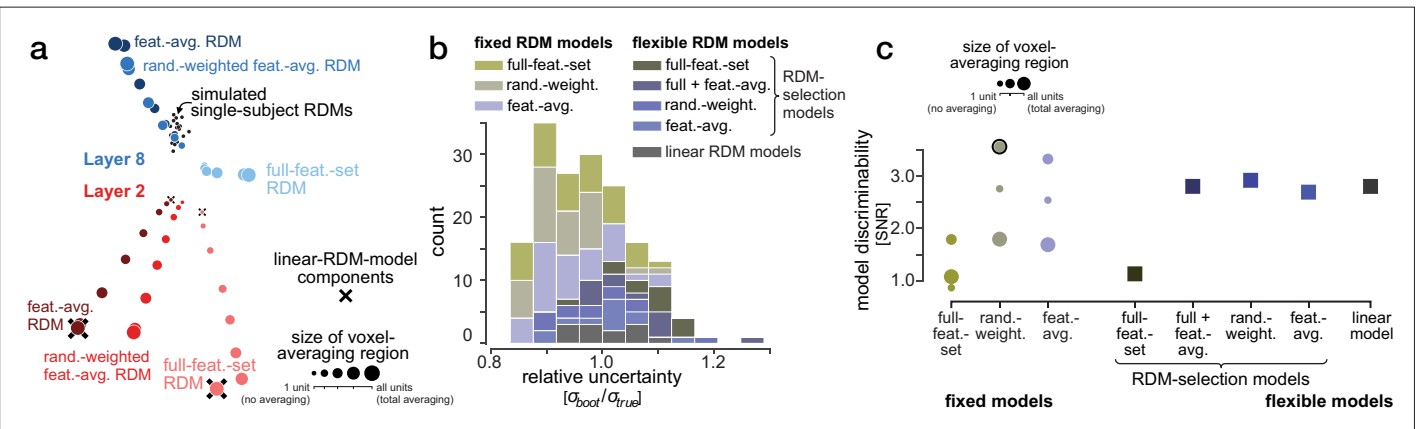

**Figure 5.** Validation of flexible model tests using bootstrap crossvalidation. (**a**) MDS arrangement of the representational dissimilarity matrices (RDMs) for one simulated dataset. Colored circles show the predictions based on one correct and one wrong layer changing the voxel averaging region and the treatment of features ('full', 'weighted', and 'avg', as described in the text). Fixed models correspond to single choice of model RDM for each layer. Selection models select the best fitting voxel size from the RDMs presented in one color (or two for 'both'). Crosses mark the four components of the linear model for Layer 2. The small black dots represent simulated subject RDMs without functional magnetic resonance imaging (fMRI) noise. (**b**) Histogram of relative uncertainties $\sigma_{boot}/\sigma_{true}$, showing that the bootstrap-wrapped crossvalidation accurately estimates the variance of the performance estimates across many different inference scenarios. (**c**) Model discriminability as signal-to-noise ratios for different model types.

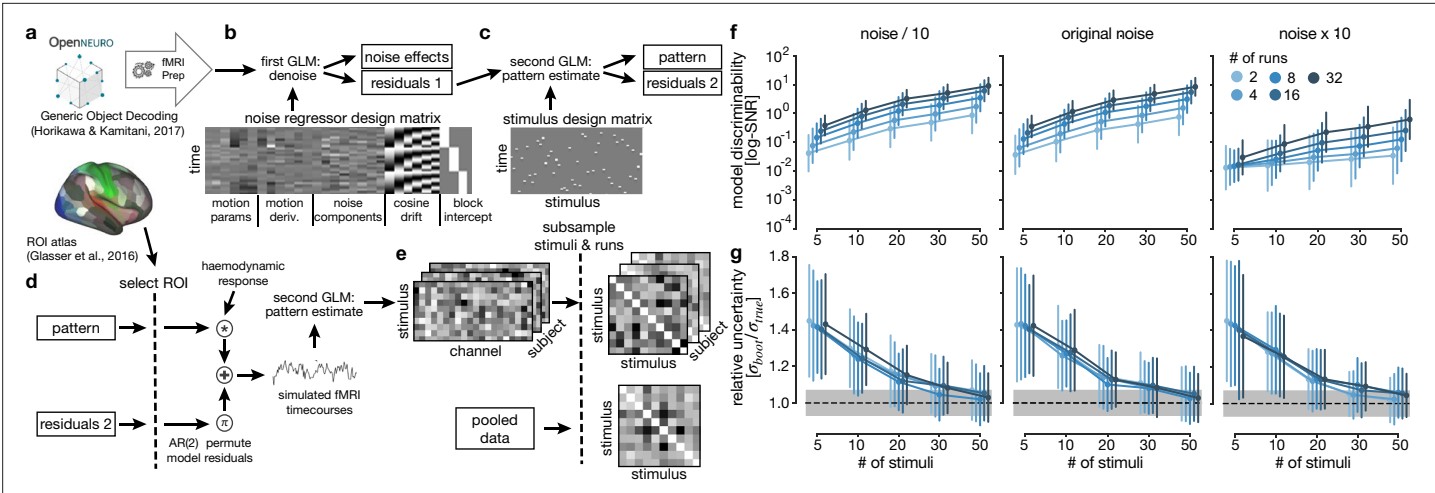

**Figure 6.** Functional magnetic resonance imaging (fMRI)-data-based simulation. (**a**) These simulations are based on a dataset of neural recordings for 50 stimuli in 5 human observers (*Horikawa and Kamitani, 2017*), which were each shown 35 times. To extract stimulus responses from these data we perform two general linear model (GLM) steps as in the original publication. (**b**) In the first step, we regress out diverse noise estimators (provided by fMRIprep) from pooled fMRI runs. (**c**) We then apply a second GLM separately on each run to extract the stimulus responses. (**d**) We then extract regions of interest (ROIs) based on an atlas (*Glasser et al., 2016*), randomly chose differently sized subsets of runs resp. stimuli to enter further analyses. To simulate realistic noise, we estimate an AR(2) model on the second GLM's residuals, permute and filter them to keep their original autocorrelation structure, and finally scale them by the factors 0.1, 1, and 10. To generate simulated timecourses, we add these altered residuals to the GLM prediction. We then rerun the second GLM on the simulated data and use the Beta-coefficient maps for following steps. (**e**) Finally, we compute crossnobis representational dissimilarity matrices (RDMs) and perform RSA based on the overall RDM across all subjects. (**f**) Results of the simulations, separately for each noise scaling factor. The signal-to-noise ratio shows the same increase as for our abstract simulation. (**g**) The relative uncertainty converges to 1 for increasing stimulus numbers. Error bars indicate standard deviations across different simulation types.

(weighted feature treatment and voxel size 0.05), but the model variants which allowed for some fitting still yield high SNRs. Analyses that take the averaging across space and features into account yielded the highest average model performance for the true model. In contrast, analyses that ignore averaging over space or features (the full feature set selection model and some of the fixed models) not only lead to lower SNRs (as seen in *Figure 5c*), but also systematically selected the wrong layer, because a higher average performance was achieved by a different layer than the one we used for generating the data (not shown).

We conclude that when the true voxel sampling is unknown, flexible models are needed to account for voxel sampling, so as to enable us recover the underlying data-generating computational model with our model-comparative inference. Fixed models based on incorrect assumptions about the voxel sampling can lead to low model-performance discriminability (SNR) and even to incorrect inferences as to which model is the true model.

## Validation with functional MRI data

The simulations presented so far validated all statistical inference procedures, but may not capture all aspects of the structure of real measurements of brain activity. To test our methods under realistic conditions, we used real human fMRI (this section) and mouse calcium-imaging data (next section). We resampled data from a large openly available fMRI experiment in which humans viewed pictures from ImageNet (*Horikawa and Kamitani, 2017*). These data contain various noise sources, individual differences, signal shapes, and distributions that are difficult to simulate accurately without using measured data. We therefore implemented a data-based simulation to create realistic synthetic data, whose ground-truth RDM we knew (*Figure 6*). By subsampling from this dataset, we generated smaller datasets to test inference with bootstrapping over conditions. We used the entire dataset as a stand-in for the population a researcher might wish to generalize to. For each cortical area, we computed the mean RDM using all data (all runs and subjects). Each area's mean RDM served as a ground-truth RDM for datasets sampled from that area and as a model RDM for datasets sampled from all areas. The model comparison we attempted aims to recover which cortical area a dataset was

subsampled from. The simulation enables us to check whether our uncertainty estimates are correct for model-performance estimates based on real data.

We varied the strength of noise, the number of runs, and the number of conditions (i.e. viewed images). We did not vary the number of subjects because the original dataset contains only five subjects, which precludes informative resampling of subjects. To increase the variability of the resampled datasets beyond sampling from the 35 measurement runs and to vary the noise strength, we created new voxel timecourses for each sampled run while preserving the spatial structure and serial autocorrelation of the noise. To achieve this, we estimated a second-order autoregressive model ($AR(2)$) separately for each run's GLM residuals, permuted the AR-model's residuals and added the results to the GLM's predicted timecourse (see *Figure 6a–e* and Methods, fMRI-data-based simulation for details). We repeated each simulated experiment 24 times and used the RU and the model-performance discriminability (SNR) as our evaluation criteria.

Results were largely similar to those of the neural-network-based simulations (*Figure 6f, g*). For the RU, which measures the accuracy of our bootstrap variance estimates, we see a convergence toward the expected ratio (dashed line at 1), validating the bootstrap procedure for real fMRI data. For the model-performance discriminability (SNR), we find the same power-law increase with the number of conditions and the number of runs used as data. These results suggest that the regions are discriminable on the basis of their RDMs estimated from fMRI given five subjects' data when a sufficient number of stimuli (≥30) and runs (≥16) is used.

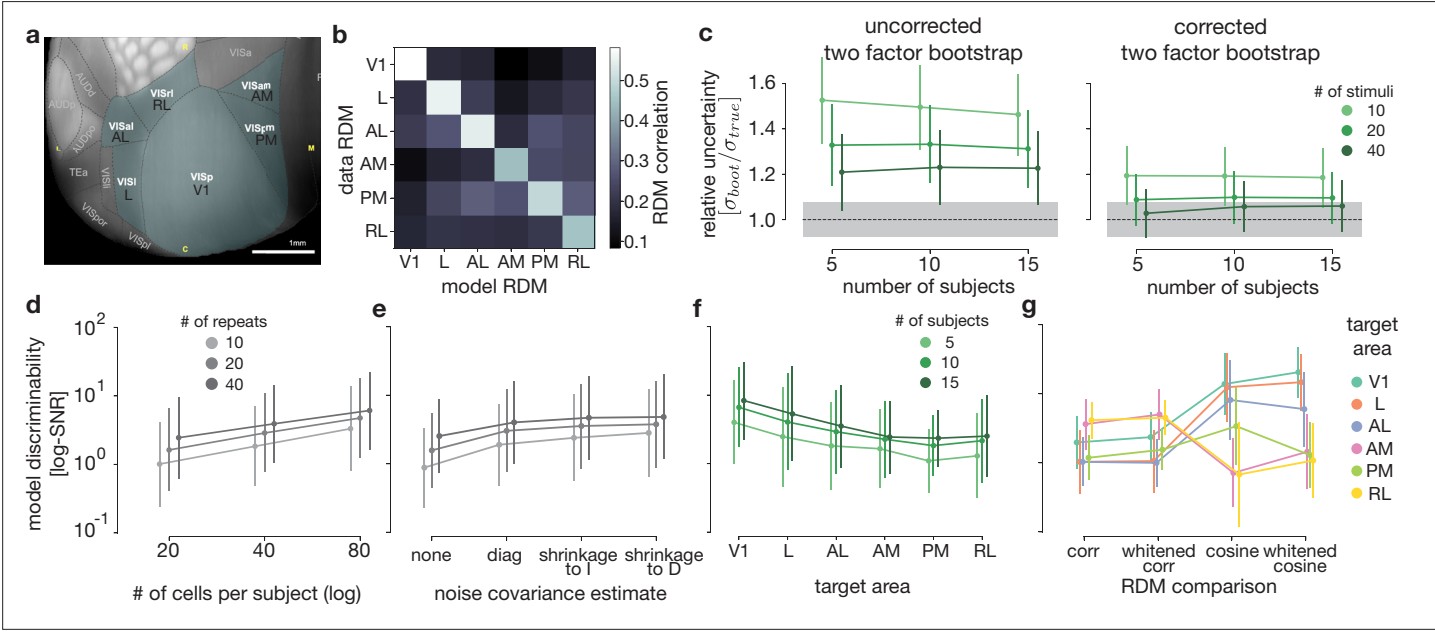

**Figure 7.** Results in mice with calcium-imaging data. (**a**) Mouse visual cortex areas used for analyses and resampling simulations. (**b**) Overall similarities of the representations in different cortical areas in terms of their representational dissimilarity matrix (RDM) correlations. For each mouse and cortical area ('data RDM', vertical), the RDM was correlated with the average RDM across all other mice ('model RDM', horizontal), for each other cortical area. We plot the average across mice of the crossvalidated RDM correlation (leave-one-mouse-out crossvalidation). The prominent diagonal shows the replicability across mice and the distinctness between cortical areas of the representational geometries. (**c**) Relative uncertainty for the 2-factor bootstrap methods. The gray box indicates the range of results expected from simulation variability if the bootstrap estimates were perfectly accurate. The correction is clearly advantageous here although the method is still slightly conservative (overestimating the true standard deviation $\sigma_{true}$ of model-performance evaluations) for small numbers of stimuli. For 40 or more stimuli, the corrected 2-factor bootstrap correctly estimates the variance of model-performance evaluations. (**d**) Signal-to-noise ratio validation: The signal-to-noise ratio (SNR) grows with the number of cells per subject and the number of repeats per stimulus. (**e**) Signal-to-noise ratio for different noise covariance estimates. Taking a diagonal covariance estimate into account, that is normalizing cell responses by their standard deviation is clearly advantageous. The shrinkage estimates provide a marginal improvement over that. (**f**) Signal-to-noise ratio for data sampled from different areas. (**g**) Which measure is optimal for discriminating the models depends on the data-generating area. On average there is an advantage of the cosine similarity over the RDM correlation and of the whitened measures over the unwhitened ones. Error bars indicate standard deviations across different simulation types.

Image credit: Allen Institute.

## Validation with calcium-imaging data

We can also adjudicate among models of the representational geometry on the basis of direct neural measurements, such as electrophysiological recordings or calcium-imaging data. These measurement modalities have very different statistical properties than fMRI. To test our methods for this kind of data, we performed a resampling simulation based on a large calcium-imaging dataset of responses of mouse visual cortex to natural images (*de Vries et al., 2020*). This dataset contains recordings from six visual cortical areas: primary visual cortex (V1), laterointermediate (LM), posteromedial (PM), rostrolateral (RL), anteromedial (AM), and anterolateral (AL) visual area (*Figure 7a*).

As in the previous section, we used the overall mean RDM for each area as a ground-truth model and subsampled the data to create simulated datasets for which we know the ground-truth RDM. We used different numbers of stimulus repetitions, neurons, mice, and stimuli to vary the amount of information afforded by each simulated dataset. We used the crossnobis estimator of representational dissimilarity for all analyses here. We repeated each simulated experiment 100 times and computed the RU to assess the correctness of our bootstrap uncertainty estimates and the model-discriminability SNR to determine which noise covariance estimators and RDM comparators afford most sensitivity to model differences.

We analyzed the overall discriminability of the brain areas (*Figure 7b*). Although cortical areas vary in the reliability of the estimated RDMs, they can be discriminated reliably when using all data. We used the RU to assess whether our bootstrap variance estimates are correct for these data (*Figure 7c*). We resampled all factors (subjects, stimuli, runs, and cells) to generate simulted datasets. Correspondingly, the analysis used bootstrapping over both subjects and stimuli. We observed correct variance estimates for the corrected 2-factor bootstrap. The uncorrected 2-factor bootstrap was conservative, substantially overestimating the true variance.

To understand how the model-comparative power depends on experimental parameters and analysis choices, we analyzed the model-discriminability SNR. We found that more subjects, more stimuli, more runs, and more cells all increased the SNR just as in our fMRI and neural-network-based simulations (*Figure 7d*). Furthermore, we find that taking the noise covariance into account for computing the crossnobis RDMs in the first-level analysis improves the SNR (*Figure 7e*). Univariate noise normalization (implemented by using a diagonal noise covariance matrix) is better than no noise normalization. Multivariate noise normalization is slightly better than univariate noise normalization (*Walther et al., 2016*). For multivariate noise normalization, we tested two different shrinkage estimators with different targets: a multiple of the identity and the diagonal matrix of variances. These two variants perform similarly. In addition, we find that different RDM comparators yield the best model discriminability for different cortical areas (*Figure 7g*). For some, cosine RDM similarity performs better, for others, Pearson RDM correlation performs better. The whitened RDM comparators are better on average, but there are cases where the unwhitened RDM comparators perform slightly better. Thus, it remains dependent on the concrete experiment (with a particular choice of conditions, tested models and underlying representational geometry), which RDM comparator affords the best power for model comparison (*Diedrichsen et al., 2020*).

## Discussion

We present new methods for inferential evaluation and comparison of models that predict brain representational geometries. The inference procedures enable generalization to new measurements, new subjects, and new conditions, treat flexible models correctly using crossvalidation, and work for any representational dissimilarity estimator and RDM comparator. For fixed as well as flexible models, our inference methods support all combinations of generalization: to new measurements using the same subjects and conditions, to new subjects, to new conditions, and to both new subjects and new conditions simultaneously. We validated the methods using simulated data as well as calcium-imaging and fMRI data, showing that the inferences are correct. The methods are available as part of an open-source Python toolbox (rsatoolbox.readthedocs.io).

## Generalizing to new measurements, new subjects, and/or new conditions

Inferential statistics is about generalization from the experimental random samples to the underlying populations. We must carefully consider the level of generalization, both at the stage of designing our experiments and analyses and at the stage of interpreting the results. The lowest level of inferential generalization is to new measurements. Our conclusions in this scenario are expected to hold only for replications of the experiment in the same animals using the same conditions. Inferential generalization to new subjects may not be possible, for example, in case studies or when the number of animals (e.g. two macaques) is insufficient. Generalization to new conditions is not needed when all conditions relevant to our claims have been sampled. For example, *Ejaz et al., 2015* studied the representational similarity of finger movements in primary motor cortex. All five fingers were sampled in the experiments and there are no other fingers to generalize to. When generalizing to replications with the same subjects and conditions, we need separate data partitions to estimate the variability of the model-performance estimates. We can then use a $t$-test or rank-sum test to test for significant differences between models.

If generalization to the population of subjects is desired, we need a sufficiently large sample of subjects. We can then evaluate each model for each subject and use a $t$-test or rank-sum test, treating the subjects as a random sample from a population. We showed that this method is valid, controlling false-positive rates at their nominal values in our matrix-normal simulations (Methods, Frequentist tests for model evaluation and model comparison). The variance across subjects here is a good estimate of the variance across the population of subjects. However, the interpretation of the results must be restricted to the exact set of experimental conditions used in the experiment.

We often would like our inferences to generalize to a population of conditions. For example, when evaluating computational models of vision, we are not usually interested in determining which models dominate just for the particular visual stimuli presented in our experiment. We are interested in models that dominate for a population of visual stimuli. Model-comparative inference can generalize to the population of conditions that the experimental conditions were randomly sampled from. The inference requires bootstrapping, because RDM prediction accuracy cannot be assessed for single conditions. We bootstrap-resample the conditions set and evaluate all models on each sample. This procedure correctly estimates our uncertainty about model-performance differences, and $t$-tests based on the estimated bootstrap variances provide valid frequentist inference.

If we want to generalize simultaneously across conditions and subjects, then the corrected 2-factor bootstrap approach provides accurate estimates of our uncertainty about model performances. These uncertainty estimates support valid inferential model comparisons, comparisons to the lower bound of the noise ceiling, and tests against chance performance. We expect the results to generalize to new subjects and conditions drawn from the respective populations sampled randomly in the experiment.

## Inference on fixed and flexible models

Our performance estimates for flexible models must not be biased by overfitting to measurement noise, subjects, or conditions. To avoid this bias, we use a novel 2-factor crossvalidation scheme that enables us to evaluate models' predictive accuracy when simultaneously generalizing to new subjects and/or new conditions. The 2-factor crossvalidation is nested in our 2-factor bootstrap procedure for estimating uncertainty. By using two crossvalidation cycles with different data partitionings for each bootstrap sample, we can accurately remove the excess variance introduced by crossvalidation. Our method provides a computationally efficient estimate of the variances and covariances of model-performance estimates for flexible models, which enables us to use a $t$-test to inferentially compare models to each other, to the lower bound of the noise ceiling, and to chance performance.

Our methods are fully general in that inference can be performed on any model for which the user provides a fitting and an RDM prediction method. In practice, the complexity of the models is limited by the requirement that we need to fit each model thousands of times in our bootstrap-wrapped crossvalidation scheme. Thus, we need a sufficiently fast and reliable fitting method for the model.

If fitting the model so often is not feasible or if the data RDMs do not provide sufficient constraints, one solution is to fit all models using a separate set of neural data before the inferential analyses. This approach is appropriate when many parameters are to be fitted, as is the case in nonlinear systems identification approaches as well as linear encoding models (*Wu et al., 2006*), where a large set of

neural fitting data is required. All conclusions are then conditional on the fitting data: Inference will generalize to new test data assuming models are fitted on the same fitting data. Our methods support fitting of lower-parametric models as part of the model-comparative inference. When applicable, this approach obviates the need for separate neural data for fitting and supports stronger generalization (not conditional on the neural fitting data).

## Supported tests and implications of test results

Our methods enable comparison of a model's RDM prediction performance (1) against other models, (2) against the noise ceiling, and (3) against chance performance. The first two of these tests are central to the evaluation of models. The test against chance performance is often also reported, but represents a low bar that we should expect most models to pass. In practice, RDM correlations tend to be positive even for very different representations, because physically highly similar stimuli or conditions tend to be similar in all representations. Just like a significant Pearson correlation indicates a dependency, but does not demonstrate that the dependency is linear, a significant RDM prediction result indicates the presence of stimulus information, but does not lend strong support to the particular model. We should resist interpreting significant prediction performance per se as evidence for a particular model (the single-model-significance fallacy; *Kriegeskorte and Douglas, 2019b*). Theoretical progress instead requires that each model be compared to alternative models and to the noise ceiling. An additional point to note is that the interpretation of chance performance, where the RDM comparator equals 0, depends on the chosen RDM comparator, differing, for example, between the Pearson correlation coefficient and the cosine similarity (*Diedrichsen et al., 2020*).

RDM comparators like the Pearson correlation and the cosine similarity are related to the distance correlation (*Székely et al., 2007*), a general indicator of mutual information. Like a significant distance correlation, a significant RDM correlation mainly demonstrates that there is some mutual information between the brain region in question and the model representation. For a visual representation, for example, all that is required is for the two representations to contain some shared information about the input images. In contrast to the distance correlation (and other nonnegative estimates of mutual information), however, negative RDM correlations can occur, indicating simply that pairs of stimuli close in one representation tend to be far in the other and vice versa. For any RDM, there is even a valid perfectly anti-correlated RDM (Pearson $r = -1$), which can be found by flipping the sign of all dissimilarities and adding a large enough value to make the RDM conform to the triangle inequality (which ensures the existence of an embedding of points that is consistent with the anti-correlated RDM). The existence of valid negative RDM correlations is important to the inferential methods presented here because it is required for our assumption of symmetric ($t$-)distributions around the true RDM correlation.

Omnibus tests for the presence of information about the experimental conditions in a brain region have been introduced in previous studies (e.g. *Kriegeskorte et al., 2006*; *Allefeld et al., 2016*; *Nili et al., 2020*). Whether stimulus information is present in a region is closely related to the question whether the noise ceiling is significantly larger than 0, indicating RDM replicability. Such tests can sensitively detect small amounts of information in the measured activity patterns and can be helpful to assess whether there is any signal for model comparisons. If we are uncertain whether there is a reliable representational geometry to be explained, we need not bother with model comparisons.

The question whether an individual dissimilarity is significantly larger than zero is equivalent to the question whether the distinction between the two conditions can be decoded from the brain activity. Decoding analyses can be used for this purpose (*Naselaris et al., 2011*; *Hebart et al., 2014*; *Tong and Pratte, 2012*; *Kriegeskorte and Douglas, 2019b*). Such tests require care because the discriminability of two conditions cannot be systematically negative (*Allefeld et al., 2016*). This is in contrast to comparisons between RDMs, which can be systematically negative (although, as mentioned above, they tend to be positive in practice).

## How many subjects, conditions, repetitions, and measurement channels?

Statistical inference gains power when more data are collected along any dimension. More independent measurement channels, more subjects, more conditions, and more repetitions all help. How much data is needed along each of these dimensions depends on the experiment. The most helpful

dimension to extend is the one that currently limits generalization. When crossvalidation across repeated measurements is used to eliminate the bias of the distance estimates (as in the crossnobis estimator), using more repetitions brings an additional performance bonus because it reduces the variance increase associated with unbiased estimates (*Diedrichsen et al., 2020*, Appendix 5).

## Which distance estimator and RDM comparator?

The statistical inference procedures introduced here work for any choice of representational-distance estimator and RDM comparator. However, the choice of distance estimator and RDM comparator affects the power of model-comparative inference and the meaning of the inferential results.

For computing the RDM, we tested only variations of the crossnobis (crossvalidated Mahalanobis) distance estimator, as recommended based on earlier research (*Walther et al., 2016*). The crossnobis estimator can use different noise covariance estimates to normalize patterns, such that the noise distribution becomes approximately isotropic. The noise covariance matrix can be the identity (no normalization), diagonal (univariate normalization), or a full estimate (multivariate normalization). Consistent with previous findings (*Walther et al., 2016*; *Ritchie et al., 2021*), our results suggest that univariate noise normalization is always preferable to no normalization, and that multivariate noise normalization using a shrinkage estimate of the noise covariance (*Ledoit and Wolf, 2004*; *Schäfer and Strimmer, 2005*) helps in some circumstances and never hurts model discrimination.

For evaluating RDM predictions, we can distinguish RDM comparison methods by the scale they assume for the distance estimates: ordinal, interval, or ratio. For ordinal comparisons, the different rank correlation coefficients perform similarly. We recommend $\rho_a$ for its computational efficiency and analytically derived noise ceiling. For interval- and ratio-scale comparisons, a more complex pattern emerges. In particular whether cosine similarities (ratio scale) or Pearson correlations (interval scale) work better depends on the structure of the model RDMs to be compared. We recently proposed whitened variants of the cosine similarity and Pearson correlation, which take into account that the distance estimates in an RDM are not independent (*Diedrichsen et al., 2020*). The whitened RDM comparators were more sensitive to subtle differences in model performance when evaluated on fixed models (*Figure 5c*). In the simulations based on the calcium-imaging data, whitened RDM comparators still performed better on average, but there were some cortical areas that were easier to identify by using the unwhitened comparison measures.

## Alternative approaches

We present a frequentist inference methodology that uses crossvalidation to obtain point estimates of model performance and bootstrapping to estimate our uncertainty about them. Bayesian alternatives deserve consideration. For example, a Bayesian approach has been proposed to alleviate the bias of distance estimates (*Cai et al., 2019*). This Bayesian estimate makes more detailed assumptions about the trial dependencies than our crossvalidated distance estimators, which remove the bias. The Bayesian estimate might be preferable for its higher stability when its assumptions hold and could be used in combination with our model-comparative inference methods. For model comparisons, Bayesian inference is also an interesting alternative to the frequentist methods we discuss here (*Kriegeskorte and Diedrichsen, 2016*). Our whitened RDM comparison methods can be motivated as approximations to the likelihood for a model and we reported recently that they afford similar power as likelihood-based inference with normal assumptions (*Diedrichsen et al., 2020*). Thus, frequentist inference using the whitened RDM comparators is related to Bayesian inference with a uniform prior across models. In the Bayesian framework, generalization to the populations of subjects and conditions would require a model of how RDMs vary across subjects and conditions. We currently do not have such a model. Until such models and Bayesian inference procedures for them are developed, the frequentist methods we present here remain the only method for generalization to the populations of subjects and conditions.

Another strongly related method for comparing models to data in terms of their geometry is pattern component modeling (*Diedrichsen et al., 2018*), which compares conditions in terms of their covariance over measurement channels instead of their representational dissimilarities. This approach is deeply related to representational similarity analysis (*Diedrichsen and Kriegeskorte, 2017*). Pattern component modeling is somewhat more rigid than RSA as the theory is based on normal distributions, but it has advantages in terms of analytical solutions. In particular, the likelihood of models can be

directly evaluated, enabling tests based on the likelihood ratio. Due to the direct evaluation of likelihoods, this framework can be combined with Bayesian inference more easily and recently a variational Bayesian analysis was presented for this model (*Friston et al., 2019*).

Another powerful approach to inference on brain-computational models is to fit encoding models that predict measured brain-activity data instead of representational geometries (e.g. *Wu et al., 2006*; *Kay et al., 2008*; *Dumoulin and Wandell, 2008*; *Naselaris et al., 2011*; *Wandell and Winawer, 2015*; *Diedrichsen and Kriegeskorte, 2017*; *Cadena et al., 2019a*). This approach was originally developed in the context of low-dimensional models and measurements. When models and measurements are both high dimensional, even a linear encoding model can be severely under-constrained (*Cadena et al., 2019b*; *Kornblith et al., 2019*). As a result, an encoding model requires a combination of substantial fitting data and strong priors on the weights. The predictive model that is being evaluated comprises the encoding model and the priors on its weights (*Diedrichsen and Kriegeskorte, 2017*), which complicates the interpretation of the results (*Cadena et al., 2019b*; *Kriegeskorte and Douglas, 2019b*). Both model performances and the fitted weights can then be highly uncertain and/or dependent on the details of the assumed encoding model. The additional data and assumptions needed to fit complex encoding models motivate the consideration of methods as proposed here that do not require fitting of a high-parametric mapping from model to measured brain activity.

The generalization challenges that we tackle here for RSA apply equally to encoding models and pattern component modeling. Inferences are often meant to generalize to new subjects and/or experimental conditions. The alternative approaches, in their current implementations, do not yet enable simultaneous generalization to the populations of experimental conditions and subjects. By default pattern component modeling and its Bayesian variants assume a single geometry and thus do not take either subject or condition variability into account. Variability across subjects can be taken into account in a group-level analysis (see e.g. *Diedrichsen et al., 2018*, 2.7.3), but this approach does not account for uncertainty due to the sample of experimental conditions. Encoding models usually follow the machine learning approach with training, validation, and test sets (e.g. *Naselaris et al., 2011*; *Cichy et al., 2019*; *Cichy et al., 2021*). Uncertainty about the model evaluations is either not estimated at all or estimated in a secondary analysis based on the variability across subjects, cells, or conditions. Because these secondary analysis is based solely on the test set, results are conditional on the training and validation sets, and so fall short of generalizing model-comparative inferences to the underlying populations. Note that the bootstrapping and crossvalidation approaches we introduce here are not inherently specific to RSA. These methods could be adapted for estimating the uncertainties about other model evaluation measures such as those provided by pattern component and encoding models.

## Conclusion

We present a comprehensive new methodology for inference on models of representational geometries that is more powerful than previous approaches, can handle flexible models, and enables neuroscientists to draw conclusions that generalize to new subjects and conditions. The validity of the methods has been established through extensive simulations and using real neural data. These methods enable neuroscientists working with humans and animals to evaluate complex brain-computational models with measurements of neural population activity. As we enter the age of big models and big data, we hope these methods will help connect computational theory to neuroscientific experiment.

## Materials and methods

The methods section for this paper is separated into two parts: First, we describe the RSA analysis pipeline we propose in full. In the second part, we describe the simulation methods we used to test our pipeline for this paper.

### Full description of the RSA method

The inference method we describe here represents a new pipeline for representational similarity analysis. Nonetheless, some parts of the analysis appeared in earlier or concurrent publications (*Kriegeskorte et al., 2008b*; *Nili et al., 2014*; *Walther et al., 2016*; *Storrs et al., 2014*). In this

section, we describe the whole pipeline, including both new and established procedures, without requiring familiarity with previous papers.

## Computing representational dissimilarity matrices

The first step of RSA is the computation of the representational dissimilarity matrices. The main question for this step is which dissimilarity measure shall be computed between conditions.

In the formal mathematical sense, a distance or metric is a function that takes two points from the space as input and computes a real number from them conforming to the following three rules: (1) *Nonnegativity*: The result is larger than or equal to zero, with equality only if the two points are equal. (2) *Symmetry*: The result is the same if the two points are swapped. (3) *Triangle inequality*: The sum of distances from $a$ to $b$ and $b$ to $c$ is less than or equal to the distance from $a$ to $c$ for all choices of the three points. We use the term dissimilarity for symmetric measures that may violate (1) and (3). Dropping requirement (3) admits symmetric divergences between probability distributions, for example. Dropping requirement (1) and allowing measures that may return negative values admits unbiased distance estimators (whose distribution is symmetric about 0 when the true distance is 0). We would still like the dissimilarity to be nonnegative in expectation.

In principle, any dissimilarity measure on the measured representation vectors can be used to quantify the dissimilarities between conditions. Popular choices in the past were the Pearson correlation distance, squared and unsquared Euclidean distances, cosine distance, and linear-decoding-based measures such as the decoding accuracy, the linear-discriminant contrast (LDC, also known as the crossnobis estimator; *Walther et al., 2016*) and the linear-discriminant $t$ value (LD-$t$; *Kriegeskorte et al., 2007*; *Nili et al., 2014*). Earlier publications comparing different measures of dissimilarity found correlation distances to be less interpretable in terms of condition decodability and continuous crossvalidated decoding measures (LDC, LD-$t$) to be more sensitive than decoding accuracy (*Walther et al., 2016*).

How representational dissimilarity is best quantified and inferred from raw data depends on the type of measurements taken. For fMRI for example, it is beneficial to take the noise covariance across voxels into account by computing Mahalanobis distances (*Walther et al., 2016*). For electrophysiological recordings of individual neurons one should take the Poisson nature of the variability into account. One approach is to transform the measured spike rates so as to stabilize their variance (*Kriegeskorte and Diedrichsen, 2019a*). Here, we introduce a KL-divergence dissimilarity measure based on the Poisson distribution (Appendix 2). Representational dissimilarities can also be inferred from behavioral responses, such as speeded categorizations or explicit judgments of properties or pairwise dissimilarities (*Kriegeskorte and Mur, 2012*).

Two aspects of the computation of dissimilarities warrant further discussion: crossvalidation of dissimilarities and taking noise covariance into account.

## Crossvalidated distance estimators

One important aspect of the first-level analysis is that naive estimates of representational similarity can be severely biased (*Walther et al., 2016*; *Cai et al., 2019*; *Diedrichsen et al., 2020*) toward a structure dictated by the structure of the experiment rather than the structure of the representations. This happens because noise in the underlying patterns biases distance estimates upward. When different conditions are measured with different amounts of noise or the measurements between some pairs of conditions are correlated, this bias will be different for different distances introducing other structure into the RDM.

To avoid this problem, one can use crossvalidated distances, which combine difference estimates from independent measurements like separate runs, such that the dissimilarity estimate is unbiased. Crossvalidation applies to all dissimilarity measures that are defined based on inner products of the differences with themselves (e.g. squared and unsquared Euclidean distances, Mahalanobis distances, Poisson KL divergences). To compute a crossvalidated dissimilarity one computes two estimates of the difference vector from independently measured parts of the data and takes the inner product of these two independent estimates, averaging over different splits into independent data.

The most commonly used version of crossvalidated distances are crossvalidated Mahalanobis (*Crossnobis*) dissimilarities, which we use througout our simulations in this paper. For $N \geq 2$ repeated

measurements of response patterns $\mathbf{x}_{mi}, m = 1 \ldots N, i = 1 \ldots K$ (for $K$ conditions), the crossnobis estimator $\hat{d}_{ij}$ is defined as:

$$\hat{d}_{ij} = \frac{1}{N(N-1)} \sum_m \sum_{n \neq m} (\mathbf{x}_{mi} - \mathbf{x}_{mj})^T \Sigma^{-1} (\mathbf{x}_{ni} - \mathbf{x}_{nj}) \tag{11}$$

for an estimate noise covariance matrix $\Sigma$.

Crossnobis dissimilarities seem to be the most reliable dissimilarity estimate for fMRI-like data (**Walther et al., 2016**). As in the non-crossvalidated Mahalanobis distance, the linear transformation of of the response dimensions (using the noise precision matrix $\Sigma^{-1}$) improves reliability (**Walther et al., 2016**; **Nili et al., 2014**) and renders the estimates monotonically related to the linear decoding accuracy for each pair of conditions, when a fixed Gaussian error distribution is assumed. Their sampling distributions can be described analytically (**Diedrichsen et al., 2020**).

For Poisson distributed measurements as for electrophysiological recordings we can also define a crossvalidated dissimilarity based on the KL-divergence as we introduce in Appendix 2.

## Noise covariance estimation

To take the noise covariance into account (in Mahalanobis and Crossnobis dissimilarities) we first need to estimate it. To do so, we can use one of two sources of information: We either estimate the covariance based on the variation of the repeated measurements around their mean or based on the residuals of a first-level analysis which estimated the patterns from the raw data. For fMRI for example, these residuals would be the residuals of the first-level GLM. In either case, we may have relatively little data to estimate a large noise covariance matrix. Making this feasible usually requires regularization. To do so the following methods are available:

- When the estimation task is judged to be entirely impossible one can reduce the Mahalanobis and Crossnobis back to the Euclidean and crossvalidated Euclidean distances by using the identity matrix instead.
- As a univariate simplification one can estimate a diagonal matrix which only takes the variances of voxels into account.
- For estimating a full covariance one may use a shrinkage estimate, which 'shrinks' the sample covariance toward a simpler estimate of the covariance like a multiple of the identity or the diagonal of variances (**Ledoit and Wolf, 2004**; **Schäfer and Strimmer, 2005**). The amount of shrinkage used in these methods fortunately can be estimated quite accurately based on the data directly such that these methods do not require parameter adjustment.

We implemented these different methods. Overall the shrinkage estimates perform best, while the other techniques are equally good in some situations. Using the sample covariance directly is not advisable unless an unusually large amount of data exists for this estimation, in which case the shrinkage estimates converge toward the sample covariance anyway.

## Comparing RDMs

The second-level analysis is comparing a measured data RDM (for each subject) to the RDMs predicted by different models. There are various measures to compare RDMs. The right choice depends on the aspects of the data RDM that the models are meant to predict. A strict measure would be the Euclidean distance (or equivalently the mean squared error) between a model RDM and the data RDM. However, we usually cannot predict the absolute magnitude of the distances because the SNR varies between subjects and measurement sessions. Allowing an overall scaling of the RDMs leads to the cosine similarity between RDMs. If we additionally drop the assumption that a predicted difference of 0 corresponds to a measured dissimilarity of 0, we can use a correlation coefficient between RDMs. For the cosine similarity and Pearson correlation between RDMs we recently proposed whitened variants which take the correlations between the different entries of the RDM into account (**Diedrichsen et al., 2020**). Finally, we can drop the assumption of a linear relationship between RDMs by using rank correlations like Kendall's $\tau$ or Spearman's $\rho$. For this lowest bar for a relationship Kendall's $\tau_a$ or randomized rank breaking for Spearman's $\rho_a$ are usually preferred over a standard Spearman's $\rho$ or Kendall's $\tau_b$ and $\tau_c$, which all favor RDMs with tied ranks (**Nili et al., 2014**). As we discuss below there

is a direct formula for the expected Spearman's $\rho$ under random tiebraking, which we prefer now for computational efficiency reasons.

## Estimating the uncertainty of our model-performance estimates

Additional to the point estimate of model performances we want to estimate how certain we should be about our evaluations. In the frequentist framework, this corresponds to an estimate how variable our evaluation results would be if we repeated the experiment. All variances we need can be computed from the covariance matrix of the model-performance estimates. Thus, we keep an estimate of this matrix as our overall uncertainty estimate.

### Subject variance

The easiest to estimate variance is the variance our results would have if we repeated the experiment with new subjects, but the same conditions, as all our evaluations are simple averages across subjects. Thus, an estimate of the variance can always be obtained by dividing the sample variance over subjects by the number of subjects.

### Bootstrapping conditions

The dependence of the evaluation on the conditions is more complicated. Thus, we use bootstrapping (**Efron and Tibshirani, 1994**) to estimate the variance we expect over repetitions of the experiment with new conditions but the same subjects. To do so, we sample sets of conditions with replacement from the set of measured conditions and generate the data RDM and the model RDMs for this sample of conditions. The variance of the model performances on these resampled RDMs is then an estimate of the variance over experiments with different stimulus choices. The bootstrap samples of conditions contain repetitions of the same condition. The dissimilarity between a stimulus and itself will be 0 in the data and any model such that every model would correctly predict these self-dissimilarities. Thus, including these self-dissimilarities would bias all model performances upward. To avoid this, we exclude them from the evaluation.

### 2-Factor bootstrap

If we want to estimate the variance for generalization to new subjects and new stimuli simultaneously, we need to use the correction we introduce in the results section (see Estimating the variance of model-performance estimates for generalization to new subjects and conditions). This yields an estimate of the model evaluation (co-)variances as all other methods for variance estimation.

As we mention in the main text, the exact formulas for the correction contain factors that depend on the number of subjects $N_s$ and conditions $N_c$, respectively. The expected value for the uncorrected 2-factor bootstrap variance $\hat{\sigma}_{sc}$ is:

$$\mathbb{E}(\hat{\sigma}_{sc}^2) = \frac{N_s - 1}{N_s}\sigma_s^2 + \frac{N_c - 1}{N_c}\sigma_c^2 + \left(\frac{(N_s - 1)(N_c - 1)}{N_s N_c} + \frac{N_s - 1}{N_s} + \frac{N_c - 1}{N_c}\right)\sigma_{sc}^2 \tag{12}$$

If the true variances due to the subject is $\sigma_s^2$, the one due to the conditions is $\sigma_c^2$, and the one due to measurement noise or interaction of the two is $\sigma_{sc}^2$. Correspondingly, the expectations for the two 1-factor bootstraps $\hat{\sigma}_s^2$ and $\hat{\sigma}_c^2$ are:

$$\mathbb{E}(\hat{\sigma}_s^2) = \frac{N_s - 1}{N_s}\sigma_s^2 + \frac{N_s - 1}{N_s}\sigma_{sc}^2 \tag{13}$$

$$\mathbb{E}(\hat{\sigma}_c^2) = \frac{N_c - 1}{N_c}\sigma_c^2 + \frac{N_c - 1}{N_c}\sigma_{sc}^2 \tag{14}$$

Combining these equations an unbiased estimate of the variance $\hat{\sigma}^2$ can be obtained:

$$\hat{\sigma}_{c2f}^2 = \frac{N_s}{N_s - 1}\hat{\sigma}_s^2 + \frac{N_c}{N_c - 1}\hat{\sigma}_c^2 - \frac{N_s N_c}{(N_s - 1)(N_c - 1)}(\hat{\sigma}_{sc}^2 - \hat{\sigma}_s^2 - \hat{\sigma}_c^2) \tag{15}$$

As $N_s$ and $N_c$ grow, the three ratio factors converge to 1, and the result converges to the one given by the simpler formula in the main text (**Equation 5**).

## Bootstrap-wrapped crossvalidation

If we employ any flexible models, we should additionally use crossvalidation, which leads to our new bootstrap-wrapped crossvalidation explained in the results section (Evaluating the performance of flexible models).

## Frequentist tests for model evaluation and model comparison

Based on the uncertainty estimates, we construct frequentist tests to compare models to each other. The default method is a $t$-test based on bootstrap-estimated variances. There is a collection of other tests available to compare model performances against each other, to the noise ceiling, or to chance performance.

Because we base our uncertainty estimates on a bootstrap, there are two types of tests we can use for these comparisons: A percentile test based on the bootstrap samples or a $t$-test based on the estimated variances.

For the percentile test, we calculate the bootstrap distribution for the differences and then test by checking whether the difference expected under the $H_0$ (usually 0) lies within the simple percentile bootstrap confidence interval. It is possible to generate more exact confidence intervals like bias-corrected and accelerated intervals based on the bootstrap samples, which might result in better tests. In our simulations and experience with natural data, however, model performances tend to be fairly symmetrically distributed around the true value, suggesting that these corrections are unnecessary.

For the $t$-test we use the variance estimated from the bootstrap and use the number of observations minus one as the degrees of freedom. When bootstrapping across both subjects and conditions, we used the smaller number to stay conservative. This approach follows *Efron and Tibshirani, 1994*, chapter 12.4.

The $t$-test has some advantages over the percentile bootstrap (*Efron and Tibshirani, 1994*): First, precise $p$ value estimates require many bootstrap samples. Especially, when smaller $\alpha$ levels or corrections for multiple comparisons are used, the percentile bootstrap can become computationally expensive and/or unreliable. Second, for small sample sizes, the bootstrap distribution does not take the uncertainty about the variance of the distribution into account. This is a similar error as taking the standard normal instead of a $t$-distribution to define confidence intervals. Third, the bootstrap distributions are discrete, which is a bad approximation in the tails of the distribution. For example, a sample of five RDMs which are all positively related to the model is declared significantly related to the model at any $\alpha$ level, because all bootstrapped average evaluations are at least as high as the lowest individual evaluation. Fourth, for the 2-factor bootstrap and the bootstrap-wrapped crossvalidation, we can give corrections for the variance, but lack techniques to generate bootstrap samples directly.

We should also note that we expect the $t$-distribution to be a good approximation for our case: We expect fairly symmetric distributions for the differences between models and average them across subjects, which should lead to a quick convergence toward a normal distribution for the model performances and their differences.

In particular, the model performances we base our tests on are not necessarily positive, allowing us to use the same techniques for the test against 0. This is different from tests that handle the original dissimilarities, where the true distances can only be positive.

## Noise ceiling for model performance

In addition to comparing models to each other, we compare models to a noise ceiling and to chance performance. The noise ceiling provides an estimate of the performance the true (data-generating) model would achieve. A model that approaches the noise ceiling (i.e. is not significantly below the noise ceiling) cannot be statistically rejected. We would need more data to reveal any remaining shortcomings of the model. The noise ceiling is not 1, because even the true group RDM would not perfectly predict all subjects' RDMs because of the intersubject variability and noise affecting the RDM estimates. We estimate an upper and a lower bound for the true model's performance (*Nili et al., 2014*). The upper bound is constructed by computing the RDM which performs best among all possible RDMs. Obviously, no model can perform better than this best RDM, so it provides a true upper bound. To estimate a lower bound, we use leave-one-out crossvalidation, computing the best performing RDM for all but one of the subjects and evaluating on the held-out subject. We can understand the upper and lower bound of the noise ceiling as uncrossvalidated and crossvalidated

estimates of the performance of an overly flexible model that contains the true model. The uncrossvalidated estimate is expected to be higher than the true model's performance because it is overfitted. The crossvalidated estimate is expected to be lower than the true model's performance because it is compromised by the noise and subject-sampling variability in the data.

For most RDM comparators, the best performing RDM can be derived analytically as a mean after adequate normalization of the single subject RDMs. For cosine similarity, they are normalized to unit norm. For Pearson correlation, the RDM vectors are normalized to zero mean and unit standard deviation. For the whitened measures the normalization is based on the norm induced by the noise precision instead, that is subject RDM vectors $\mathbf{d}$ are divided by $\sqrt{\mathbf{d}^T \Sigma^{-1} \mathbf{d}}$ instead of the standard Euclidean norm $\sqrt{\mathbf{d}^T \mathbf{d}}$. For the Spearman correlation, subject RDM vectors are first transformed to ranks.

For Kendall's $\tau_a$, there is no efficient method to find the optimal RDM for a dataset, which is one of the reasons for using the Spearman rank correlation for RDM comparisons. If Kendall $\tau$ based inference is chosen nonetheless, the problem can be solved approximately by applying techniques for Kemeny–Young voting (*Ali and Meilă, 2012*) or by simply using the average ranks, which is a reasonable approximation, especially if the rank transformed RDMs are similar across subjects. In the toolbox, we currently use this approximation without further adjustment.

For the lower bound, we use leave-one-out crossvalidation over subjects. To do this, each subject is once selected as the left-out subject and the best RDM to fit all other subjects is computed. The expected average performance of this RDM is a lower bound on the true model's performance, because fitting all distances independently is technically a very flexible model, which performs the same generalization as the tested models. As all other models it should thus perform worse than or equal to the correct model.

When flexible models are used, such that crossvalidation over conditions is performed, the computation of noise ceilings needs to take this into account (*Storrs et al., 2014*). Essentially, the computation of the noise ceilings is then restricted to the test sets of the crossvalidation, which takes into account which parts of the RDMs are used for evaluation.

## Flexible models

As model types, we implement three types of flexible model, in addition to the standard fixed model, which represents a single RDM to be tested:

1. A *selection model*, which states that one of a set of RDMs is the correct one.
2. A one-dimensional *manifold model*, which consists of an ordered list of RDMs and is allowed to linearly interpolate between neighboring RDMs.
3. A *weighted representational model*, which states that the RDM is a (positively) weighted sum of a set of RDMs.

These models aim to provide the flexibility necessary to appropriately represent the uncertainty about the data generation process in different ways. First, we may be uncertain about aspects of the underlying brain-computational model, such as the relative prevalence in the neural population of different subpopulations of tuned neurons (*Khaligh-Razavi and Kriegeskorte, 2014*; *Khaligh-Razavi et al., 2017*; *Jozwik et al., 2016*). Second, we may be uncertain about aspects of the measurement process, such as the spatial smoothing and weighted averaging of features due to measurement methods. Measurement with functional MRI voxels, for example, can strongly influence the resulting RDMs, which can lead to wrong conclusions and generally bad model performance when the models are compared to measured RDMs (*Kriegeskorte and Diedrichsen, 2016*).

The selection model implements flexibility in perhaps the simplest way, by allowing a choice among a set of RDMs produced from the model under different assumptions about the brain computations and/or the measurement process. For training, we can simply evaluate each possible RDM on the training data and choose the best performing one as the model prediction for evaluation. This model implies no structure in which RDMs can be predicted, but can only handle a finite set of RDMs.

The one-dimensional manifold model implements an ordered set of RDMs, where the model is allowed to interpolate between each pair of consecutive RDMs. This representation is helpful if the uncertainty about the measurements effect can be well summarized by one continuous parameter like the width of a smoothing kernel. Then we can sample a set of values for this parameter and use the simulated results as the basis RDMs for this kind of model. Then the model will provide an

approximation to the continuous set of RDMs predicted by changing the parameter without requiring a method to optimize the parameter directly.

Finally, the weighted representational model represents the effect of weighting orthogonal features. In this case, the overall RDM is a weighted sum of the RDMs generated by the individual features. Whenever the original model has a feature representation, we may be uncertain about the features relative prevalence in the neural population and/or in the measured responses. It can be sensible then to assume that these features are represented with different weights or are differently amplified by the measurement process. The squared Euclidean representational distances that would obtain from a concatenation of feature subsets, each multiplied by a different weight, is equal to a nonnegatively weighted combination of the squared Euclidean RDMs for the individual feature sets. This justifies a nonnegative weighted model at the level of the RDMs.

A particular application of the weighted representation model is motivated by the local averaging in fMRI voxels, which leads to an overrepresentation of the population-mean dimension of the multivariate response space (*Carlin and Kriegeskorte, 2017*; *Kriegeskorte and Diedrichsen, 2016*; *Ramírez et al., 2014*). The expected RDM for measurements that independently randomly weight features is a linear combination of two RDMs, one based on treating features separately and one based on averaging all features before computing the RDM (see Appendix 4).

Our methodology is not specific to these types of model and can be easily extended to other types of model. To do so, the only requirement is that there is a reasonably efficient fitting method to infer the best fitting parameters for a given dataset of training RDMs. Indeed, new model types can be slotted into our toolbox by users by implementing only two functions: one that predicts an RDM based on a parameter vector and one that fits the parameter vector to a dataset.

## Validation of the methodology

To evaluate our methods we use three kinds of simulations. First, we implement simulations based on deep neural networks and a simple approximation of voxel sampling. By choosing a new random voxel sampling per subject and using different randomly selected input images, we can test our methods with systematic variations across conditions and/or subjects. Second, we implement a simulation based on real fMRI data recombining measurements signals and noise to keep all complications found in true fMRI data. Third, we present simulations based on calcium-imaging data from mice (*de Vries et al., 2020*).

Additionally, we tested that the tests we implement are in principle valid using a simple simulation based on a normal distribution for the original measurements, which corresponds to the matrix-normal generative model we used for theoretical derivations elsewhere (*Diedrichsen et al., 2020*). These simulations are presented in Appendix 1.

### Neural-network-based simulation

Our simulations were based on the activities in the convolutional layers of AlexNet (*Krizhevsky et al., 2012*) in response to randomly chosen images from the ecoset validation set (*Mehrer et al., 2021*). For each stimulus, we computed the activities in the convolutional layer and took randomly chosen local averages to simulate the averaging of voxels. We then generated fMRI-like measurement timecourses to a randomly ordered short event-related design by convolution with a hemodynamic-response function and addition of autoregressive noise. We then ran a GLM analysis to estimate the response strength to each stimulus. From these estimated voxel responses, we computed data RDMs per subject and ran our proposed analysis procedures to compute model performances of different models which we also based on the convolutional layers of AlexNet.

For the network, we used the implementation available for pytorch through the torchvision package (*Paszke et al., 2019*).

### Stimuli

Stimuli were chosen independently from the validation set of ecoset by first choosing a category randomly and then sampling an image randomly from that category. These stimuli are natural images with categories chosen to approximate the relevance for human observers. The validation set contain 565 categories with 50 images each, that is 28,250 images in total.

### Noise-free voxel response

To compute the response strength of a voxel to a stimulus we computed a local average of the feature maps. We first convolved the feature maps with a Gaussian representing the spatial extend of the voxels, whose size we defined by its standard deviation relative to the overall size of the feature map. A voxel with size 0.05 would thus correspond to a Gaussian averaging area whose standard deviation is 5% of the size of the feature map. Voxel locations were then chosen uniformly randomly over the locations within the feature map. To average across features, we chose a weight for each feature and each voxel uniformly between 0 and 1 and then took the weighed sum as the voxel response.

### fMRI simulation

To generate timecourses we assumed a measurement was taken every 2 s and a new stimulus was presented during every second measurement, with no stimulus presented in the measurement intervals between stimulus presentations.

To generate a simulated fMRI response, we computed the stimulus by voxel response matrix and normalized it per subject to have equal averaged squared value. We then converted this into timecourses following the usual GLM assumptions and convolved the predictions with a hemodynamic-response function. We set the hrf to the standard sum of two gamma distributions as assumed in statistical parametric mapping (SPM; *Pedregosa et al., 2015*), normalized to an overall sum of 1.

We then added noise from an autoregressive model of rank 1 (AR1) with covariance between pairs of voxels given by the overlap of the weighting functions of their weights. To control the strength of the autocorrelation, we set the coefficient for the previous data point to 0.5. To enforce the covariance between voxels, we multiplied the noise matrix with the cholesky decomposition of the desired covariance. To control the overall noise strength we scaled the final noise by a constant.

Each stimulus was presented once per run, with multiple stimulus presentations implemented as multiple runs.

### Analysis

To analyse the simulated data we ran a standard GLM analysis which yielded a $\beta$-estimate for each presented stimulus for each run of the experiment.

To compute RDMs we used Crossnobis distances based on leave-one-out crossvalidation over runs and the covariance of the residuals of the GLM. For this step, we used the function implemented in our toolbox.

### Fixed-model definition

As models to be compared we used the different layers of AlexNet. To generate an optimal model RDM we applied two transformations to mimic the average effect of voxel sampling. First we convolved the representation with the spatial receptive field of the voxels to mimic the spatial averaging effect. To capture the effect of pooling the features with nonnegative weights, we then computed a weighted sum of the RDM containing the features separately and one RDM based on the summed response across features weighted with weights 1 and 3.

This weighting computes the expected Euclidean distance of patterns under our random weighting scheme as we derive in Appendix 4: For our $w_i \sim U(0, 1)$ the expected value is $\mathbb{E}(w) = \frac{1}{2}$ and the variance is $\mathrm{Var}(w_i) = \frac{1}{12}$ such that the weights for the RDM based on the individual features is $\frac{1}{4}$ and the weight for the RDM based on the summed feature response is $\frac{1}{12}$, that is a 3:1 weighting.

Based on this weighting we generated a fixed model for each individual processing step in AlexNet including the nonlinearities and pooling operations resulting in 12 models predicting a fixed RDM.

### Tested conditions

For the large deep-neural-network-based simulation underlying the results in *Figure 4*, we chose a base set of factors which we crossed with all other conditions and a separate set of factors which were not crossed with each other but only with the base set.

Into the base set of factors we included the following factors: Which experimental parameters were changed over repetitions of the experiment (none, subjects, conditions, or both) and which bootstrapping method we applied (over conditions, over subjects, over both or applying the bootstrap correction). We applied all four bootstrapping conditions to the simulations in which none of the parameters

were varied, the fitting ones to the subject and stimulus varying simulations and the bootstrap with and without correction for to the simulations were both parameters varied over repetitions resulting in 8 conditions for variation and bootstrap. Additionally, we included the number of subjects (5, 10, 20, 40, or 80) and the number of conditions (10, 20, 40, 80, and 160). For each set of conditions we thus ran 8 × 5 × 5 = 200 conditions.

Other factors we varied were: The number of repeats, which we set to 4 usually and tested 2 and 8. The layer we used to simulate the data, which we usually set to layer number 8 which corresponds to the output of the 3rd convolutional layer, and also tried 2, 5, 10, and 12, which correspond to the other 4 convolutional layers of AlexNet. The size of the voxels which we usually set to 0.05, that is we set the standard deviation of the Gaussian to 5% of the size of the feature map. As variations we tried 0, 0.25, and ∞, that is no spatial pooling, a quarter of the size of the feature map as standard deviation and an average over the whole feature map. Finally, we varied the number of voxels, which we usually set to 100, but tried 10 and 1000 additionally. In total we thus ran 3 + 5 + 4 + 3 = 15 sets of conditions with 200 conditions each resulting in 3000 conditions, with a grand total of 300,000 simulations.

### Bootstrap-wrapped crossvalidation

To test the precision and consistency of the calculations for the bootstrap-wrapped crossvalidation (*Figure 5a, b*), we needed repeated analyses for the same datasets. For this simulations we thus simulated only 10 datasets for the standard conditions, 20 subjects and 40 conditions, while varying both conditions and subjects and then ran repeated analyses on these datasets. For each setting, we ran 100 repeated analysis of each dataset. As conditions we chose 2, 4, 8, 16, and 32 crossvalidation assignments for 1000 bootstrap samples and additionally variants with only 2 crossvalidation folds and 2000, 4000, 8000, or 16,000 bootstrap samples.

### Flexible-model treatment

To test whether our methods are adequate for estimating the variability for model performances of flexible models (*Figure 5d–f*), we ran our standard settings for 20 subjects and 40 stimuli and drawing new subjects and new stimuli, while replacing the fixed models per layer with flexible models of different kinds.

We generated models by combining models with different assumptions about the voxel pooling pattern: We varied two factors: (1) How feature weighting was handled: full, that is predicted distances are Euclidean distances in the original feature space; avg, that is distances are the differences in the average activation across features; or 'weighted', that is the weighted average of these two models, that corresponds to the expected RDM under the weight sampling we simulated. (2) How averaging over space was handled.

We first used different kinds fixed models, which serve as the building blocks for the flexible models. We varied two aspects of the measurement models applied: How large voxels are assumed to be (no pooling, $std = 5\%$ of the image size and pooling over the whole feature maps) and how the features pooling is handled (no pooling, average feature, or the correct weighting assumed for the fixed models previously). These 3 × 3 combinations are the 9 fixed-model variants.

We then generated selection models which had a range of voxel sizes to choose from (no pooling, std = 1%, 2%, 5%, 10%, 20%, 50% of the image size and pooling across the whole feature map). For the treatment of pooling over features we used four variants: For the first three called full, average, and weighted we used one of the types of fixed models to generate the RDMS. For the last, we allowed both the RDMs used by the full models and the ones used by the average models as a choice.

As an example of a linearly weighted model, we generated a model which was allowed to use a linear weighting of the four corner-case RDMs: no feature pooling and no spatial pooling, average feature and no spatial pooling, a global average per feature map, and the RDM induced by pooling over all locations and features. The model could predict any linear combination of the corner-case RDMs to fit the data RDMs.

### fMRI-data-based simulation

With our fMRI-data-based simulation, we aim to show that our analyses are correct and functional for real fMRI data. Real data may contain additional statistical regularities, which we did not take into account in our deep-neural-network-based simulations. To do so, we took a large published dataset

of fMRI responses to images and sampled from this dataset to generate hypothetical experimental datasets across which we would like to generalize. All scripts for the fMRI-data-based simulation are openly available on https://github.com/adkipnis/fmri-simulations, (*Kipnis, 2023*).

## Dataset

For these simulations we used data from *Horikawa and Kamitani, 2017* (as available from https://openneuro.org/datasets/ds001246/versions/1.2.1). This dataset contains fMRI data collected from five subjects viewing natural images selected from ImageNet or imagining images from a category. For our simulations, we used only the 'test' datasets, which contain 50 different images from distinct categories, which were each presented 35 times to each subject giving us an overall reliable signal and repetitions to resample from.

We used the automatic MRI preprocessing pipeline implemented in fMRIPrep 1.5.2 (*Esteban et al., 2019*; *Esteban et al., 2019*; RRID:SCR_016216), which is based on *Nipype* 1.3.1 (*Gorgolewski et al., 2011*; *Gorgolewski et al., 2018*; RRID:SCR_002502). This program was also used to produce the following description of the preprocesing performed.

## Anatomical-data preprocessing

The T1-weighted (T1w) image was corrected for intensity non-uniformity (INU) with N4BiasFieldCorrection (*Tustison et al., 2010*), distributed with ANTs 2.2.0 (*Avants et al., 2008*, RRID:SCR_004757), and used as T1w reference throughout the workflow. The T1w reference was then skull stripped with a *Nipype* implementation of the antsBrainExtraction.sh workflow (from ANTs), using OASIS30ANTs as target template. Brain tissue segmentation of cerebrospinal fluid (CSF), white matter (WM), and gray matter (GM) was performed on the brain-extracted T1w using fast (FSL 5.0.9, RRID:SCR_002823, *Zhang et al., 2001*). Brain surfaces were reconstructed using recon-all (FreeSurfer 6.0.1, RRID:SCR_001847, *Dale et al., 1999*), and the brain mask estimated previously was refined with a custom variation of the method to reconcile ANTs- and FreeSurfer-derived segmentations of the cortical GM of Mindboggle (RRID:SCR_002438, *Klein et al., 2017*). Volume-based spatial normalization to one standard space (MNI152NLin2009cAsym) was performed through nonlinear registration with antsRegistration (ANTs 2.2.0), using brain-extracted versions of both T1w reference and the T1w template. The following template was selected for spatial normalization: *ICBM 152 Nonlinear Asymmetrical template version 2009c* (*Fonov et al., 2009*, RRID:SCR_008796; TemplateFlow ID: MNI152NLin2009cAsym).

## Functional data preprocessing

For each of the 35 blood-oxygen-level-dependent (BOLD) runs found per subject (across all tasks and sessions), the following preprocessing was performed. First, a reference volume and its skull-stripped version were generated using a custom methodology of *fMRIPrep*. The BOLD reference was then co-registered to the T1w reference using bbregister (FreeSurfer) which implements boundary-based registration (*Greve and Fischl, 2009*). Co-registration was configured with six degrees of freedom. Head-motion parameters with respect to the BOLD reference (transformation matrices, and six corresponding rotation and translation parameters) are estimated before any spatiotemporal filtering using mcflirt (FSL 5.0.9, *Jenkinson et al., 2002*). BOLD runs were slice-time corrected using 3dTshift from AFNI 20160207 (*Cox and Hyde, 1997*, RRID:SCR_005927). The BOLD time-series, were resampled to surfaces on the following spaces: *fsaverage5* and *fsaverage6*. The BOLD time-series (including slice-timing correction when applied) were resampled onto their original, native space by applying the transforms to correct for head-motion. These resampled BOLD time-series will be referred to as *preprocessed BOLD in original space*, or just *preprocessed BOLD*. The BOLD time-series were resampled into standard space, generating a *preprocessed BOLD run in ['MNI152NLin2009cAsym'] space*. First, a reference volume and its skull-stripped version were generated using a custom methodology of *fMRIPrep*. Several confounding time-series were calculated based on the *preprocessed BOLD*: framewise displacement (FD), the spatial root mean square of the data after temporal differencing (DVARS) and three region-wise global signals. FD and DVARS are calculated for each functional run, both using their implementations in *Nipype* (following the definitions by *Power et al., 2014*). The three global signals are extracted within the CSF, the WM, and the whole-brain masks. Additionally, a set of physiological regressors were extracted to allow for component-based noise correction (*Behzadi et al., 2007*). Principal components are estimated after high-pass filtering the *preprocessed*

*BOLD* time-series (using a discrete cosine filter with 128 s cut-off) for the two *CompCor* variants: temporal (tCompCor) and anatomical (aCompCor). tCompCor components are then calculated from the top 5% variable voxels within a mask covering the subcortical regions. This subcortical mask is obtained by heavily eroding the brain mask, which ensures it does not include cortical GM regions. For aCompCor, components are calculated within the intersection of the aforementioned mask and the union of CSF and WM masks calculated in T1w space, after their projection to the native space of each functional run (using the inverse BOLD-to-T1w transformation). Components are also calculated separately within the WM and CSF masks. For each CompCor decomposition, the *k* components with the largest singular values are retained, such that the retained components' time-series are sufficient to explain 50% of variance across the nuisance mask (CSF, WM, combined, or temporal). The remaining components are dropped from consideration. The head-motion estimates calculated in the correction step were also placed within the corresponding confounds file. The confound time-series derived from head-motion estimates and global signals were expanded with the inclusion of temporal derivatives and quadratic terms for each (*Satterthwaite et al., 2013*). Frames that exceeded a threshold of 0.5 mm FD or 1.5 standardized DVARS were annotated as motion outliers. All resamplings can be performed with *a single interpolation step* by composing all the pertinent transformations (i.e. head-motion transform matrices, susceptibility distortion correction when available, and co-registrations to anatomical and output spaces). Gridded (volumetric) resamplings were performed using antsApplyTransforms (ANTs), configured with Lanczos interpolation to minimize the smoothing effects of other kernels (*Lanczos, 1964*). Non-gridded (surface) resamplings were performed using mri_vol2surf (FreeSurfer).

Many internal operations of *fMRIPrep* use *Nilearn* 0.5.2 (*Abraham et al., 2014*, RRID:SCR_001362), mostly within the functional processing workflow. For more details of the pipeline, see the section corresponding to workflows in *fMRIPrep*'s documentation.

The above boilerplate text was automatically generated by fMRIPrep with the express intention that users should copy and paste this text into their manuscripts *unchanged*. It is released under the CC0 license.

## Region selection

Visual areas were defined according to the surface atlas by *Glasser et al., 2016*. For our simulations we used the following 10 visual areas as ROIs, joining areas from the atlas to avoid too small ROIs (the name of the areas in the atlas is given in brackets): V1 (V1), V2 (V2), V3 (V3), V4 (V4), ventral visual complex (VVC), ventromedial visual area (VMV1, VMV2, VMV3), parahippocampal place area (PHA1, PHA2, PHA3), fusiform face area (FFC), inferotemporal cortex (TF, PeEc), and MT/MST (MT, MST). The areas were selected separately for the two hemispheres.

To map the atlas onto individual subject's brain space we used the mappings estimated by FreeSurfer with fmriprep's standard settings. The Glasser Atlas was registered to each participant's native space with mri_surf2surf, and voxels labeled using mri_annotation2label. Next, each ROI was mapped to native T1w volumetric space with mri_label2vol. To cover as many contiguous voxels as possible, the resulting masks were inflated with mri_binarize and every voxel outside of the volume between the pial surface and WM was eroded with mris_calc. To convert the resulting masks to T2*w space we used custom python scripts: First, masks were smoothed with a Gaussian kernel of FWHM = 3 mm, resampled to T2*w space using nearest neighbor interpolation, and finally thresholded. The threshold for each mask was set to equalize mask volume between T1w and T2*w space. Finally, voxels with multiple ROI assignments were removed from all ROIs but the one with the highest pre-threshold value. Voxels outside the fMRIPrep-generated brain mask were removed from all generated 3d-masks of ROIs.

## General linear modeling

For extracting response patterns from the measurements we used two GLMs. In the first GLM, we regressed out noise sources and in the second we estimate stimulus responses. This two-step process is advantageous in this case where stimulus predictors and noise predictors are highly collinear (As there is only one presentation of each stimulus per run and as they cover the whole run they can together form almost any sufficiently slow variation.): It allows us to attribute all variance that could be attributed to the noise sources uniquely to them and not to effects of stimulus presentation. This

is closer to the original papers analysis, leads to higher reliability and generates the second GLM as a stage at which we can adequately model the noise with a relatively simple AR(2) model.

GLM was performed in SPM12 (http://www.fil.ion.ucl.ac.uk/spm). No spatial smoothing was applied, models were estimated using Restricted Maximum Likelihood on top of Ordinary Least Squares and auto-correlations were taken into account using SPM's inbuilt AR(1) method.

For the first GLM, we used the following noise regressors from the ones provided by fMRIprep: An intercept for each run, the six basic motion parameters and their derivatives, six cosine basis functions to model drift, FD, DVARS, and the first six aCompCors with the largerst eigenvalues. All runs were pooled to get the best noise parameter estimates possible.

We interpret the residuals from the first GLM as a denoised version of the fMRI signal and use them as input for a second GLM separately for each run to estimate stimulus effects: Stimulus-specific regressors were generated by convolving stimulus onset time-series with the canonical HRF without derivatives. From this GLM, we kept the estimated $\beta$ coefficients $\hat{\beta}_i \in \mathbb{R}^p$ for each stimulus and the residuals $r_i$ for further processing.

### Resampling

To sample a single run for further analysis, we randomly chose a run from the measured data without replacement. To expand the set of possible datasets, we then generated a new simulated BOLD signal $\tilde{y}_i$ for each voxel $i$ at the stage of the second-level GLM. To do so, we model the data as a GLM with an AR(2) model for the noise and then generate a new timecourse by permuting the residuals $\eta_i$ of the AR(2) model. As we apply the same permutation to each voxel, this procedure largely preserves spatial noise covariance.

In mathematical formulas, this process can be described as follows: Let $p$ be the number of conditions, $n$ be the number of scans per run, and $y_i \in \mathbb{R}^n$ be the denoised BOLD response of voxel $i$. We can then use the design matrix of the run $X \in \mathbb{R}^{n \times p}$, the point estimate $\hat{\beta}_i \in \mathbb{R}^p$ for the parameter values and corresponding residuals $r_i \in \mathbb{R}^n$ estimated by SPM to simulate a new data run:

$$\tilde{y}_i = X\hat{\beta}_i + \lambda \cdot \tilde{r}_i, \qquad \tilde{r}_{i,t} = \hat{w}_{i,1}\tilde{r}_{i,t-1} + \hat{w}_{i,2}\tilde{r}_{i,t-2} + \tilde{\eta}_{i,t} \qquad (16)$$

where $w_{i,1}$ and $w_{i,2}$ are the estimated parameters of an AR(2) model fitted to the residuals $\mathbf{r}_i$. Its residuals are denoted $\eta_i$ and were randomly permuted to give $\tilde{\eta}_i$ using the same permutation for all voxels in a run.

Additionally, we sampled the conditions to use with replacement, that is we used the $\beta_i$ of a random sample of conditons, which we also used to select the RDMs from the models.

We saved this dataset in the same format as the original data and re-ran the second-level GLM using SPM on these simulated data to generate noisy estimates stimulus responses in each voxel.

### RDM calculation and comparison

We use crossnobis RDMs for this simulation, testing four different estimates for the noise covariance: We either use the identity, univariate noise normalization, or a shrinkage estimate of the covariance based on the covariance of the residuals, or based on the covariance of the individual runs' mean-centered $\beta$ estimates.

For comparing RDMs, we use the cosine similarity throughout.

### Model RDMs

We use the RDMs of different ROIs as models, effectively testing how well our methods recover the data-generating ROI. The model RDM for each ROI is the pooled RDM across all subjects and runs computed by the same noise normalization method as the one used for the data RDMs. Data for these RDMs stemmed from the original results of the second-level GLM, making them less noisy than any RDM stemming from the simulated data.

## Simulation design

For each condition, we ran 100 repeats to estimate the true variability of results and ran all combinations of the following conditions: We used 2, 4, 8, 16, or 32 runs per simulation (5 variants). We used 5, 10, 20, 30, or 50 stimuli (5 variants). We scaled the noise by 0.1, 1.0, or 10.0 (3 variants). We used each of the 20 ROIs for data generation once (20 variants). And we used the 4 methods for estimating

the noise covariance (4 variants). Resulting in 24· 5· 5· 3· 20·4 = 144,000 different simulations. To save computation time, we ran the fMRI simulation and analyses only once per repeat and noise level and ran all analysis variants on the same data. When fewer runs or conditions were required for a variant, we randomly selected a subset for the analysis without replacement.

### Calcium-imaging-data-based simulation

For the calcium-imaging-data-based simulation we used the Allen institutes mouse visual coding calcium-imaging data available at https://observatory.brain-map.org/visualcoding/ (*de Vries et al., 2020*). Detailed information on the recording techniques can be obtained from the original publications and with the dataset.

We used the 'natural scenes' data, which consists of measured calcium responses to 118 natural scenes. The natural scenes were shown for 250 ms each without an inter stimulus interval in random order. In each session, each image was present 50 times.

From this dataset, we selected all experimental sessions, which contained a natural scenes experiment. Additionally, we restricted ourselves to three relatively broad cre driver lines, which target excitatory neurons relatively broadly: 'Cux2-CreERT2', 'Emx1-IRES-Cre', and 'Slc17a7-IRES2-Cre'. For further analyses, we ignore which driver line was used to achieve enough data for resampling. This resulted in 174 experimental sessions from 91 mice with 146 cells recorded on average (range: 18–359). Of these recordings, 35 came from laterointermediate area, 32 from posteromedial visual area, 23 from rostrolateral visual area, 46 from primary visual cortex, 16 from anteromedial area, and 22 from anterolateral area.

To quantify the response of a neuron to the stimuli, we used the fully preprocessed $\frac{df}{F}$ traces as provided by the dataset. We then extract the measurements from the frame after the one marked as stimulus onset till the stated stimulus endframe resulting in six or seven frames per stimulus presentation. As a response per neuron we then simply took the average of these frames.

To compute RDMs based on these data, we used Crossnobis distances based on different estimates of the noise covariance matrix based on the variance of the stimulus repetitions around the average neural response for each stimulus. We either used: an identity matrix, effectively calculating a crossvalidated Euclidean distance; a diagonal matrix of variances, corresponding to univariate noise normalization; a shrinkage estimate toward a constant diagonal matrix (*Ledoit and Wolf, 2004*), or a shrinkage estimate shrunk toward the diagonal of sample variances (*Schäfer and Strimmer, 2005*).

To generate new datasets, we randomly sampled subsets of stimuli, mice, runs, and cells from a brain area without replacement. To exclude possible interactions we avoided sampling multiple sessions recorded from the same mouse by sampling the mice and then randomly sampling from the sessions of each mouse, if there were more than one. For this dataset, we did not use any further processing of the data.

As variants for this simulation, we performed all combinations of the following factors: 20, 40, or 80 cells per experiment; 5, 10, or 15 mice; 10, 20, or 40 stimuli; 10, 20, or 40 stimulus repeats; the four types of noise covariance estimates; four types of rdm comparison: cosine similarity, correlation, whitened cosine similarity, and whitened correlation; whether the bootstrap was corrected; and the six brain areas. This resulted in 3 × 3 × 3 × 3 × 4 × 4 × 2 × 6 = 15,552 simulation conditions for which we simulated 100 simulations each.

As models for the simulations, we used the average RDM for each brain area as a fixed RDM model for that brain area. Thus the models are not independent from the data in our main simulation. This is not problematic for checking the integrity of our inference methods, but does not show that we can indeed differentiate brain areas based on their RDMs. To show that retrieving the brain area is possible as displayed in *Figure 7b* in the main text, we performed leave-one-out crossvalidation across mice, that is we chose the RDM models for the brain areas based on all but one mice and evaluated the RDM correlation with the left out mouse's RDM.

## Additional information

### Competing interests

Jörn Diedrichsen: Reviewing editor, *eLife*. The other authors declare that no competing interests exist.

## Funding

| Funder | Grant reference number | Author |
|--------|------------------------|--------|
| Deutsche Forschungsgemeinschaft | Forschungsstipendium SCHU 3351/1-1 | Heiko H Schütt |

The funders had no role in study design, data collection, and interpretation, or the decision to submit the work for publication.

## Author contributions

Heiko H Schütt, Conceptualization, Data curation, Software, Formal analysis, Supervision, Funding acquisition, Validation, Investigation, Visualization, Methodology, Writing – original draft, Writing – review and editing; Alexander D Kipnis, Data curation, Software, Formal analysis, Validation, Investigation, Visualization, Writing – original draft, Writing – review and editing; Jörn Diedrichsen, Conceptualization, Software, Formal analysis, Supervision, Methodology, Writing – review and editing; Nikolaus Kriegeskorte, Conceptualization, Resources, Data curation, Software, Formal analysis, Supervision, Funding acquisition, Investigation, Visualization, Writing – original draft, Project administration, Writing – review and editing

## Author ORCIDs

Heiko H Schütt (iD) https://orcid.org/0000-0002-2491-5710
Jörn Diedrichsen (iD) http://orcid.org/0000-0003-0264-8532
Nikolaus Kriegeskorte (iD) http://orcid.org/0000-0001-7433-9005

## Decision letter and Author response

Decision letter https://doi.org/10.7554/eLife.82566.sa1
Author response https://doi.org/10.7554/eLife.82566.sa2

# Additional files

## Supplementary files

• MDAR checklist

## Data availability

No new data were collected for this study. The code to run both the analysis we do and our simulations is available with our rsatoolbox (https://github.com/rsagroup/rsatoolbox, copy archived at *Schütt, 2023*). The data for the fMRI-based resampling analysis are available from *Horikawa and Kamitani, 2017* and the data for the calcium imaging are available from the Allen institutes website.

The following previously published datasets were used:

| Author(s) | Year | Dataset title | Dataset URL | Database and Identifier |
|-----------|------|---------------|-------------|-------------------------|
| Horikawa T, Kamitani Y | 2017 | Generic decoding of seen and imagined objects using hierarchical visual features | https://openneuro.org/datasets/ds001246 | Open Neuro, ds001246 |

*Continued on next page*

*Continued*

| Author(s) | Year | Dataset title | Dataset URL | Database and Identifier |
|---|---|---|---|---|
| de Vries SEJ, Lecoq JA, Buice MA, Groblewski PA, Ocker GK, Oliver M, Feng D, Cain N, Ledochowitsch P, Millman D, Roll K, Garrett M, Keenan T, Kuan L, Mihalas S, Olsen S, Thompson C, Wakeman W, Waters J, Williams D, Barber C, Berbesque N, Blanchard B, Bowles N, Caldejon SD, Casal L, Cho A, Cross S, Dang C, Dolbeare T, Edwards M, Galbraith J, Gaudreault N, Gilbert TL, Griffin F, Hargrave P, Howard R, Huang L, Jewell S, Keller N, Knoblich U, Larkin JD, Larsen R, Lau C, Lee E, Lee F, Leon A, Li L, Long F, Luviano J, Mace K, Nguyen T, Perkins J, Robertson M, Seid S, Shea-Brown E, Shi J, Sjoquist N, Slaughterbeck C, Sullivan D, Valenza R, White C, Williford A, Witten DM, Zhuang J, Zeng H, Farrell C, Bernard A, Phillips JW, Reid RC, Koch C | 2020 | A large-scale standardized physiological survey reveals functional organization of the mouse visual cortex | https://observatory.brain-map.org/visualcoding/ | Allen Brain Atlas, visualcoding |

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

## Appendix 1

### Matrix-normal simulations

To establish the validity of our model-comparative frequentist inference, we need to look at the false-alarm rate for data that is generated under the assumption that the null hypothesis $H_0$ is true, that is that a model has chance performance in expectation or that two models, predict distinct RDMs, but achieve equal RDM prediction accuracy in expectation. In the deep-neural-network-based simulations and the data-resampling simulations in the main text, we are not able to generate such data. Here, we used a matrix-normal model, as a simpler simulation scheme for RSA in which we can enforce these null hypotheses.

We started by specifying a desired RDM for our data that fulfills the null hypothesis for the model(s). We then exploit the relationship between the RDM and the covariance matrix between conditions (*Diedrichsen and Kriegeskorte, 2017*) to find a covariance matrix that results in the given RDM and generate responses with this covariance between conditions. This random pattern will then have the desired (squared Euclidean) RDM.

Concretely, the second-moment matrix $\mathbf{G}$ of inner products among condition-related patterns across voxels can be computed from the squared Euclidean-distance matrix $\mathbf{D}$ as follows:

$$\mathbf{G} = -\frac{1}{2}(\mathbf{HDH}) \tag{17}$$

where $\mathbf{H} = \mathbf{I}_k - \frac{1}{K}\mathbf{1}_k$ is a centering matrix with $\mathbf{1}_k$ being a square matrix of ones. A dataset with this covariance across conditions has $\mathbf{D}$ as its squared Euclidean RDM (*Diedrichsen and Kriegeskorte, 2017*). We can easily generate Gaussian data with a given second-moment matrix and can thus generate data with any desired RDM.

### Comparison against 0

To generate $H_0$ data for testing our comparisons of models against 0, we choose both the model and the data RDMs as the distances between independent drawn Gaussian noise samples.

### Model comparisons

To generate $H_0$ data for model comparisons, we first generate two random model RDMs from independent standard normal noise data. We then normalize the model RDMs to have 0 mean and standard deviation of 1. Then we average the two RDMs, which yields a matrix with equal correlation to the two models. As a last step, we then subtract the minimum, to yield only positive distances and add the maximum distance to all distances once, such that the triangle inequality is guaranteed. As this last step only shifted the distance vector by a constant, the final distance vector still has the exact same correlation with the two model predictions. These methods effectively draw the covariance over conditions from a standardized Wishart distribution with as many degrees of freedom as the number of measurement channels.

### Random conditions

To generate $H_0$ data for model comparisons with variance due to stimulus selection, we created two models for a large set of 1000 conditions, and generated a data RDM and covariance matrix that would yield equal performance as for the other model-comparison simulations. We then sampled a random subset of the conditions for each simulated experiment.

### Data generation

In all cases, we find a new configuration of data points that produce the desired RDM for each subject by converting the RDM into the second-moment matrix via *equation 17* and drawing random normal data as described at the beginning of this section. We then add additional i.i.d. normal 'measurement noise' to each entry of the data matrix. From this data matrix we then compute a squared Euclidean-distance RDM per subject and use this as the data RDM to enter our inference process. Finally, we run our inference methods on these data RDMs and the original model RDMs to check whether the false-positive rate matches the nominal level.

## Selected conditions

For each test and setting we generated 50 randomly drawn model RDMs and 100 datasets for each of these RDMs. We always used 200 measurement dimensions and tested all combinations of the following factors: 5, 10, 20, or 40 subjects, 5, 20, 80, or 160 conditions and all test types. As tests we used percentile tests and $t$-tests based on bootstrapping both dimensions, subjects only or conditions only, a standard $t$-test across subjects and a Wilcoxon rank-sum test. For the corrected bootstrap, we only used the $t$-test based on the estimated variances, because we cannot draw bootstrap samples based on our correction.

## Results

All test results are shown in *Appendix 1—figure 1*. They mostly turned out as expected. The classical $t$- and Wilcoxon tests performed very similar to the bootstrap tests based on subjects. For the tests against chance performance and the model comparisons with fixed conditions the false-positive rates are all close to the nominal 5%. However, we observed some inflated false-positive rates for the bootstraps at small sample sizes: About 7% when using the $t$-test and up to 12% for the simple percentile bootstrap test. These slightly too large false-positive rates are due to the bootstrap estimating the biased variance estimate (dividing by $N$ instead of $N-1$). For more than 20 subjects, we cannot distinguish the percentage from 5% anymore. For the $t$-test and Wilcoxon rank-sum test, there were no such caveats as they consistently achieved a false-positive rate of about 5%.

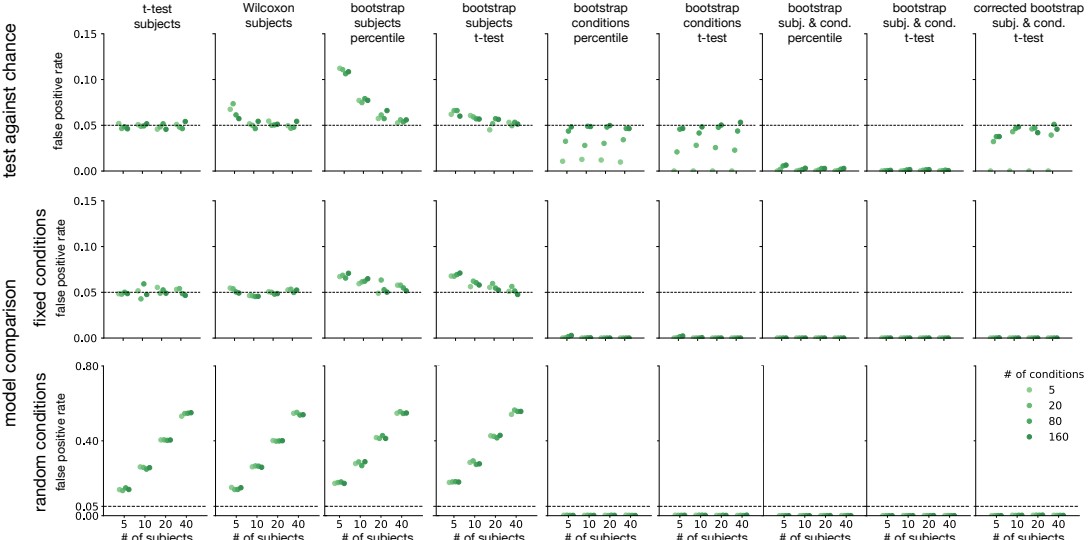

**Appendix 1—figure 1.** Evaluation of the tests using normally distributed data simulated under different null hypotheses. Each plot shows the false-positive rate plotted as a function of the number of subjects and conditions used. Ideal tests should fall on the dotted line at the nominal alpha level of 5%. Dots below the line indicate tests that are valid but conservative. Dots above the line are invalid. The 'test against chance' simulations (top row) evaluate tests of the ability of a model to predict RDMs. Data are simulated under the null hypothesis of no correlation between the data and model RDMs. A positive result would (erroneously) indicate that the model predicts the data RDM better than expected by chance. The 'model comparison' simulations (middle and bottom row) evaluate tests that compare the predictive accuracy of two models. Data are simulated under the null hypothesis that both models are equally good matches to the data. For the 'fixed conditions' simulations (middle row) this was enforced for the exact measured conditions. For the 'random conditions' simulations (bottom row) we instead generated models that are equally good on a large set of 1000 conditions, of which only a random subset of the given size is available for the inferential analysis.

Once we introduce variance due to stimulus selection by random sampling of the conditions (*Appendix 1—figure 1* third row), all methods based solely on the variance across subjects fail catastrophically with false-positive rates of up to 60% that grow with the number of tested subjects. This effect demonstrates the need to include the bootstrap across conditions into the evaluation.

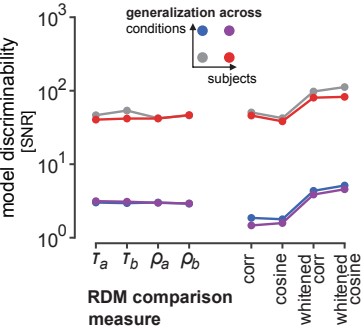

**Appendix 1—figure 2.** Sensitivity to model differences of different RDM comparators. We used the data simulated on the basis of neural network representations of images to assess how well different models (neural network layer representations) can be discriminated for model-comparative inference when using different RDM comparators. We plot the model discriminability (signal-to-noise ratio, *Equation 10*) computed for the same simulated data for each RDM comparator and generalization objective (to new measurements of the same conditions in the same subjects: gray, to new measurements of the same conditions in new subjects: red, to new measurements of new conditions in the same subjects: blue, and to new measurements of new conditions in new subjects: purple). Because the condition-related variability dominated the simulated subject-related variability here, model discriminability is markedly higher (gray, red) when no generalization across conditions is attempted. The different rank-based RDM comparators $\tau_a, \tau_b, \rho_a, \rho_b$ perform similarly and at least as well as the Pearson correlation (corr) and cosine similarity (cosine), while requiring fewer assumptions. This may motivate the use of the computationally efficient $\rho_a$, which we introduce in Appendix C. Better sensitivity to model differences can be achieved using the whitened Pearson (whitened corr) and whitened cosine similarity (whitened cosine).

When bootstrap resampling the conditions, the tests were conservative, achieving false-positive rates below 1%, lower than the nominal 5% (at the expense of power). This held whether or not subjects were treated as a random effect: The *t*-tests based on either the corrected or the uncorrected 2-factor bootstrap similarly had false-positive rates below 1%. This conservatism is expected for the tests against chance and the model comparisons with fixed conditions, because these simulations contained no true variation due to sampling of conditions. All techniques that include resampling the conditions also remain valid and conservative in the random conditions simulations that add some variation due to the condition choice. In particular, the corrected bootstrap remains conservative despite yielding strictly lower variance estimates than the uncorrected bootstrap.

We additionally ran a similar simulation, to test the tests against the noise ceiling, which is not displayed in the figure, but the results of this simulation are quickly summarized: We generated a single random model and used the same RDM also for data generation. In these data, the lower noise ceiling never significantly outperformed the true model indicating that the comparison against the lower noise ceiling is a very conservative test. This is most likely due to the difference between the lower bound on the noise ceiling and the true noise ceiling.

## Appendix 2

### Poisson KL-divergence

Instead of the Gaussian variability implied by the Euclidean and Mahalanobis dissimilarity measures, noise is often assumed to be Poisson or at least to have its variance increase linearly with mean activation. This is used primarily when the spiking variability of neurons is thought to be the main noise source as in electrophysiological recordings. For this case, we discuss two possible solutions.

The first alternative, discussed by *Kriegeskorte and Diedrichsen, 2019a* is to use a variance stabilizing transform, that is to apply a square root to all dimensions of all representations and use an RDM based on the transformed values. This has the advantage, that the covariances can be taken into account.

The second alternative, which we introduce here, is to use a symmetrized KL-divergence of Poisson distributions with mean firing rates given by the feature values. This approach automatically takes the increased variance at larger activation levels into account and inherits nice information-theoretic and decoding-based interpretations from the KL-divergence.

The KL-divergence of two Poisson distributions with mean rates $\lambda_1$ and $\lambda_2$ is given by:

$$KL(\lambda_1 \| \lambda_2) = \sum_{k=0}^{\infty} P(k|\lambda_1) \log \frac{P(k|\lambda_1)}{P(k|\lambda_2)} \tag{18}$$

$$= \lambda_1 \log \frac{\lambda_1}{\lambda_2} + \lambda_2 - \lambda_1 \tag{19}$$

Based on this we can compute the symmetrized version of the KL:

$$KL_{sym}(\lambda_1, \lambda_2) = KL(\lambda_1 \| \lambda_2) + KL(\lambda_2 \| \lambda_1) \tag{20}$$

$$= (\lambda_1 - \lambda_2)(\log \lambda_1 - \log \lambda_2) \tag{21}$$

To get a crossvalidated version of this dissimilarity we can calculate the difference in logarithms from one crossvalidation fold and the difference between raw values for a different fold and average across all pairs of different crossvalidation folds.

This KL-divergence-based dissimilarity is theoretically more interpretable than the square-root transform, but comes with two small drawbacks: First, the underlying firing rates cannot be 0 as a Poisson distribution which never fires is infinitely different from all others. This can be easily fixed by using a weak prior on the firing rate, which results in a non-zero estimated firing rate. Second, there is no straightforward way to include a noise covariance into the dissimilarity. While such noise correlations are much weaker than correlations between nearby voxels in fMRI or nearby electrodes in MEG, correlated noise may still reduce or enhance discriminability based on large neural populations (*Averbeck et al., 2006*; *Kriegeskorte and Wei, 2021*). There might be situations when the need to model noise correlations is a good reason to prefer the square-root transform.

# Appendix 3

## Spearman's $\rho_a$

*Nili et al., 2014* recommended Kendall's $\tau_a$ as the RDM comparator over other rank correlation coefficients whenever any of the models predicts tied ranks. Kendall's $\tau_a$ does not prefer predictions with tied ranks over random orderings of the same entries in expectation, making it a valid RDM comparator when any model predicts the same dissimilarity for any pair of conditions. However, Kendall's $\tau$-type correlation coefficients are considerably slower to compute than Spearman's $\rho$-type correlation coefficients. Moreover, finding the RDM with the highest average $\tau_a$ for a given set of data RDMs (for computing noise ceilings) is equivalent to the Kemeny–Young method for preference voting (*Kemeny, 1959*; *Young and Levenglick, 1978*), which is NP-hard and in practice too slow to compute for our application (*Ali and Meilă, 2012*).

Here, we propose using the expectation of Spearman's $\rho$ under random tie breaking as the RDM comparator instead. The coefficient $\rho_a$ was described by *Kendall, 1948*, chapter 3.8 and is derived below. For a vector $\mathbf{x} \in \mathbb{R}^n$, let $Rae(\mathbf{x})$ be the distribution of random-among-equals rank-transforms, where each unique value in $\mathbf{x}$ is replaced with its integer rank and, in the case of a set of tied values, a random permutation of the corresponding ranks. For each draw $\tilde{\mathbf{x}} \sim Rae(\mathbf{x})$, thus, $x_i < x_j \Rightarrow \tilde{x}_i < \tilde{x}_j$. However, for pairs $(i, j)$, where $x_i = x_j$, the ranks will fall in order $\tilde{x}_i < \tilde{x}_j$ or $\tilde{x}_i > \tilde{x}_j$ with equal probability. The set of values $\{\tilde{x}_i | 1 \leq i \leq n\}$ is $\{1, \ldots, n\}$. The $\rho_a$ correlation coefficient is defined as:

$$\rho_a(\mathbf{x}, \mathbf{y}) = \mathbb{E}_{\substack{\tilde{\mathbf{x}} \sim Rae(\mathbf{x}) \\ \tilde{\mathbf{y}} \sim Rae(\mathbf{y})}} \left[ \rho(\tilde{\mathbf{x}}, \tilde{\mathbf{y}}) \right] \tag{22}$$

For this expectation, we can derive a direct formula:

$$\begin{aligned}
\rho_a(\mathbf{x}, \mathbf{y}) \quad &= \mathbb{E}_{\substack{\tilde{\mathbf{a}} = \tilde{\mathbf{x}} - \frac{1}{n} \sum_{i=1}^{n} i, \tilde{\mathbf{x}} \sim Rae(\mathbf{x}) \\ \tilde{\mathbf{b}} = \tilde{\mathbf{y}} - \frac{1}{n} \sum_{i=1}^{n} i, \tilde{\mathbf{y}} \sim Rae(\mathbf{y})}} \left[ \frac{\tilde{\mathbf{a}}^\top \tilde{\mathbf{b}}}{\|\tilde{\mathbf{a}}\|_2 \|\tilde{\mathbf{b}}\|_2} \right] \\[1em]
&= \frac{12}{n^3 - n} \mathbb{E}_{\tilde{\mathbf{a}}}[\tilde{\mathbf{a}}]^\top \mathbb{E}_{\tilde{\mathbf{b}}}[\tilde{\mathbf{b}}] \\[1em]
&= \frac{12 \bar{\mathbf{x}}^\top \bar{\mathbf{y}}}{n^3 - n} - \frac{3(n+1)}{n - 1}
\end{aligned} \tag{23}$$

where $\bar{\mathbf{x}}$ and $\bar{\mathbf{y}}$ contain the ranks of $\mathbf{x}$ and $\mathbf{y}$, respectively, with tied values represented by tied average ranks. Thus, computing $\rho_a$ does not require drawing actual random tie breaks to sample $\tilde{\mathbf{x}}$ and $\tilde{\mathbf{y}}$.

The RDM comparator $\rho_a$ provides a general solution for rank-based evaluation that is correct in the presence of tied predictions and fast to compute. In addition, the mean of rank-transformed RDMs provides the best-fitting RDM, obviating the need for optimization and approximation in computing the noise ceiling.

## Appendix 4

### Expected RDM under random feature weighting

If measurements weight features identically and independently, we can directly compute the expected squared Euclidean RDM for the measurements. We use this calculation both to justify a linear weighting model and to compute the correct models in some of our simulations.

Formally, we can show that this is true by the following calculation: Let $w_{iv}$ be the weighting for the $i$ th feature in the $v$ th voxel for two patterns $\mathbf{x}$ and $\mathbf{y}$ with feature values $x_i$ and $y_i$. Then the expected squared Euclidean distance in voxel space can be written as:

$$\mathbb{E}\left[\frac{1}{N_v}\sum_v\left(\sum_i w_{iv}(x_i-y_i)\right)^2\right] = \mathbb{E}\left[\left(\sum_i w_i(x_i-y_i)\right)^2\right] \tag{24}$$

$$= \mathbb{E}\left[\sum_i w_i^2(x_i-y_i)^2\right] + \mathbb{E}\left[\sum_i\sum_{j\neq i} w_i w_j(x_i-y_i)(x_j-y_j)\right] \tag{25}$$

$$= \mathrm{Var}\left[w\right]\sum_i(x_i-y_i)^2 + \mathbb{E}^2\left[w\right]\sum_i\sum_j(x_i-y_i)(x_j-y_j) \tag{26}$$

$$= \mathrm{Var}\left[w\right]\sum_i(x_i-y_i)^2 + \mathbb{E}^2\left[w\right]\left(\sum_i x_i - \sum_i y_i\right)^2 \tag{27}$$

This means that the expected RDM is a linear combination of the RDM based on individual features and the RDM based on the average across features weighted by the variance and the squared expected value of the weight distribution, respectively. As averaging or filtering across space is interchangeable with feature weighting, we can also use this calculation to compute the expected RDM for models that combine averaging over space and across features as in our simulations. Then the RDM at some level of averaging over space is still always a linear combination of the feature-averaged and feature-separate RDMs at that level of spatial averaging.

# Appendix 5

## Choosing experimental design parameters for sensitive model adjudication

To quantify how much increasing the number of measurements along one of the experimental factors improved SNR for adjudication among models (*Equation 10*), we can use the slope of a regression line for the SNR against the number of measurements in log–log space. This slope corresponds to the exponent of the power-law relationship (*Figure 4* in the main text). We observe that increasing the number of conditions (*slope* = 0.935) is slightly more effective than increasing the number of subjects (*slope* = 0.690), and increasing the number of repeated measurements is most effective (*slope* = 1.581), probably due to the crossvalidation we employ. The crossvalidation across repeated measurements we use to yield unbiased distance estimates produces $\frac{m}{m-1}$ times the variance in the original RDM entries compared to the biased estimates without crossvalidation. This provides an additional benefit for increasing the number of repeated independent measurements.

The model-discriminability SNR depends on the sources of nuisance variation included in the simulations. In these particular simulated experiments, resampling the conditions set induces more nuisance variation than resampling the subjects set (*Figure 4g*). This indicates that inference generalizing across conditions is harder than inference generalizing across subjects in these simulations. For small noise levels, the SNR is much higher when nothing is varied over repetitions or only subjects are varied than when the conditions are also varied. At large measurement noise levels this effect disappears, because the measurement noise becomes the dominant factor.

The intuition to explain our observations about the SNR is that it is most helpful to take more samples along the dimension which currently causes most variation in the results. Clearly, our variation in conditions caused more variance than our variation in voxel sampling to simulate subject variability. As a result, to boost the model-discriminability SNR, increasing the number of conditions is more effective than increasing the number of subjects by the same factor. Results also reveal that we simulated sufficiently high noise levels for a reduction in measurement noise through more repetitions to remain effective. Beyond noise reduction through averaging, more repetitions are also more profitable due to the crossvalidated distances, which loses less efficiency the larger the sample becomes *Diedrichsen et al., 2020*.

Additionally, we observe that an intermediate voxel size (Gaussian kernel width) yields the highest model discriminability as measured by the SNR (*Figure 4h* in the main text). When each voxel averages over a large area, information in fine-grained patterns of activity is lost, which is detrimental to model selection. The fall-off for very small voxels in our simulations is due to randomly sampled voxels covering the feature map less well leading to greater variability. In real fMRI experiments, we do not expect this effect to play a role, as we expect voxels to always cover the whole-brain area, such that smaller voxels correspond to more voxels, which are clearly beneficial for better model selection. We do nonetheless expect a fall of for small voxel sizes for real fMRI experiments as well, because small voxel sizes lead to a steep increase in instrumental noise for fMRI and the BOLD signal itself is not perfectly local to the neurons that cause it (*Bodurka et al., 2007*; *Chaimow et al., 2018*; *Weldon and Olman, 2021*). Thus, the dependence on voxel averaging size is what we expect for real fMRI experiments as well, albeit for different reasons. Also, it might be informative for other measurement methods like electrophysiology, that a local average can be preferable over perfectly local measurements for model selection, when the number of measured channels is limited.

## Appendix 6

### Choosing an RDM comparator for sensitive model adjudication

An important question is how to measure RDM prediction accuracy for model evaluation. We ran the same analysis with different RDM comparators on the same datasets in a separate simulation.

We presented the deep neural network with our standard set of stimuli and simulated data for 10 subjects, 40 conditions, and 2 repeats, changing which parameters varied over repetitions of the experiment as in the main simulation. We omitted all bootstrapping, because the bootstrap variances are not needed to estimate model discriminability (SNR, *Equation 10*) for different RDM comparators. To improve comparability between different generalization conditions, we enforced that the first simulation for each generalization condition used the same conditions and subjects. The other 99 simulations then varied conditions and subject according to the required generalization. The different RDM comparators were applied to the same simulated experimental data.

We found that different types of rank correlation are all similarly good at discriminating models (*Appendix 1—figure 2*). Proper evaluation of models predicting tied dissimilarities requires Kendall's $\tau_a$ (*Nili et al., 2014*) or $\rho_a$, a rarely used variant of Spearman's rank correlation coefficient without correction for ties, analogous to Kendall's $\tau_a$ (derivation in Appendix 3). We recommend $\rho_a$ over $\tau_a$ for its lower computational cost and analytically derived noise ceiling.

If we are willing to assume that the representational dissimilarity estimates are on an interval scale, we expect to be able to achieve greater model-performance discriminability with RDM comparators that are not just sensitive to ranks. In this context, we compare the Pearson correlation and cosine similarity, and their whitened variants, which we introduced recently (*Diedrichsen et al., 2020*). The whitened measures boost the power of inferential model comparisons, by accounting for the anisotropic sampling distribution of RDM estimates. To further increase our model-comparative power, both the whitened and the unwhitened cosine similarity assume a ratio scale for the representational dissimilarities, which requires that indistinguishable conditions have an expected dissimilarity of zero. This assumption is justified when using a crossvalidated distance estimator (*Nili et al., 2014*; *Walther et al., 2016*), which provides unbiased dissimilarity estimates with an interpretable zero point.

Consistent with the theoretical expectations, we observe greatest model-performance discriminability for the whitened cosine similarity, which assumes ratio-scale dissimilarities, intermediate discriminability for the whitened Pearson correlation, and somewhat lower model-performance discriminability for the unwhitened Pearson correlation and the unwhitened cosine similarity. Rank correlation coefficients performed surprisingly well, matching or even outperforming unwhitened Pearson correlation and unwhitened cosine similarity (*Appendix 1—figure 2*). They provide an attractive alternative to the whitened criteria when researchers wish to make weaker assumptions about their model predictions.

