## [Editor Report]

Schütt and colleagues introduce a new method for statistical inference on representational geometries based on a cross-validated two-factor bootstrap that allows for generalization across both participants and stimuli while allowing the fitting of flexible models. In a series of elegant simulations and empirical analyses on existing datasets, the authors validate the method statistically. The work provides a fundamental and compelling advance for the analysis of representational geometries.

---

## [Decision Letter]

**Decision letter after peer review:**

Thank you for submitting your article "Statistical Inference on Representational Geometries" for consideration by *eLife*. Your article has been reviewed by 2 peer reviewers, and the evaluation has been overseen by a Reviewing Editor and Timothy Behrens as the Senior Editor. The reviewers have opted to remain anonymous.

Essential revisions:

The authors appear to get lost in details and some of the key methods are hard to find in the Methods section. This will make the paper hard to follow for those not super familiar with the analysis approach.

Expanding the section on testing for the presence of information would be very useful to broaden the appeal.

Expanding the discussion and exposition of the generalizability of the statistical tests (see Reviewer #1).

*Reviewer #1 (Recommendations for the authors):*

Schütt and colleagues introduce a new method for statistical inference on representational geometries based on a cross-validated two-factor bootstrap that allows for generalization across both participants and stimuli while allowing the fitting of flexible models. In a series of elegant simulations and empirical analyses on existing datasets, the authors validate the method statistically.

Strengths:

– The authors are clearly experts on the methods, and the statistical approach significantly improves upon the state of the art of existing methods in terms of generalization across participants and stimuli.

– There is a potential for this method to not only become a new standard for analyses of representational geometries but to be applicable to different methodological approaches that not only aim at generalizing to new participants but also to new stimuli.

– The treatment of the topic is very thorough, with both extensive simulations as well as validation using functional MRI and calcium imaging data.

– The authors introduce a number of complex yet highly informative and useful new methodological advances, such as the (re)discovery of Spearman rho_a for improved comparison of dissimilarities as compared to Kendall's tau_a.

Weaknesses:

– Overall, while the introduction starts off very nicely, the manuscript ends up being rather difficult to read. The authors appear to get lost in details in the main text. Other critical methodological details are buried in the Methods section. Specifically, the key methodological advance, the two-factor bootstrap, is barely explained in the main text, and in my reading, it is never mentioned what data are bootstrapped (i.e. original data, rows and columns in the RDM, individual cells in the RDM).

– The authors assume a lot of knowledge from the reader, often referring to very recent work or preprints in a matter-of-fact kind of way. While this can be seen as a strength and highlights the timeliness of the work, the constant mix of more established and recent methods makes it much harder for the reader to understand what is actually introduced in this work. This separation is solved nicely in the introduction but does not appear to continue into the Results section.

– Representational similarity analysis is recommended by the authors to be used for model comparison. However, a very common, probably even more common, use case is to test for the presence of information (i.e. is the representational similarity > 0), which, however, is only briefly discussed.

– The validity of the T-test based on bootstrap estimates for tests against chance seem to assume a null distribution for individual model-data comparisons that is centered around zero. However, negative similarities cannot be explained by population variance in the population null distribution, which is currently not discussed by the authors.

Are the claims of the authors justified?

– For comparisons between models, the claims of the authors clearly seem to be justified and reflect an important advance in the state-of-the-art statistical evaluation of representational geometries.

– That said, I believe that it is important to clarify the open statistical issue of generalizability to the population.

1. I really enjoyed reading this manuscript and believe it will make an important contribution to the field. That said, the authors introduce a lot in this work that is only indirectly related to the statistical analysis framework, and as a consequence, the manuscript is currently quite dense and hard to follow. I think that this manuscript would benefit strongly from a much more focused treatment of the key aspects (the introduction of the new method) and a reduction of the emphasis on advanced methods that are not key to this work (such as the use of reweighting and neuronal population sampling approaches, to name only a few).

I think the issue is that given the flexibility of analyzing representational geometries with RSA, the authors try to be as general as possible and try to encompass *all* possible use cases in their writing. In addition, the specific use case for a cross-validated two-factor bootstrap seems to be fitting flexible models, which alone is already quite advanced. I know this is difficult to solve, so I would like to provide one specific recommendation for making the manuscript easier to digest: it would perhaps help to first provide the reader with a quick run-through, without justifying all steps in detail but only summarizing the approach and the basic motivation for it. Then a more thorough treatment, including relevant parts from the methods section that explains the motivation behind the two-factor bootstrap could follow, again followed by extensive validation. This is just one suggestion for improving clarity.

2. Given the very common use of RSA for testing the presence of effects, rather than model comparison, I think the impact of this work would be strengthened if the authors expanded on their specific use case, even if it is comparably simple (they call this "simple dependence test", which is perhaps a little confusing to the reader).

3. RSA measures the match of one or more model RDMs with a data RDM. For a test against chance, without very specific biases, a negative representational similarity should not be found empirically for subjects and only for a subset of stimuli. Any such effects should thus only be caused by measurement noise or by stimulus variability. I am wondering to what degree this affects the ability to carry out valid inferences against the null at the population level. See Allefeld et al. (2016) for the treatment of a similar problem with decoding accuracies.

4. The introduction would benefit from a better motivation of the method. It seems as if the authors discuss previous work on RDMs but then jump to the introduction of the new method. Did no other method exist before? What were the issues with these methods, and what is the gap that needs to be filled? This would help the reader better understand why they should be reading this work.

5. While valid, the approach appears to be rather conservative, producing very low false positive rates. Are false negatives not a potentially problematic issue in that respect?

*Reviewer #2 (Recommendations for the authors):*

This paper addresses a major question in computational neuroscience by proposing a novel methodology to test models to explain behavioral/brain data that generalize across conditions and subjects using bootstrapping.

The experiments reported validate the claims of the authors. The methodology is applied and analysed in different available datasets.

I found particularly helpful and thorough the tests with the simulations. However, I found that the reported analysis is focused mainly on the newly proposed method, and this could bring a wider perspective into the picture.

It is with such simulated data that I believe a deeper discussion and possibly adding a comparison to existing methods, such as vanilla RSA and/or linear encoding methods could be reported to support the final discussion on the limitations of such existing methods. This would allow showcasing in which cases this method reveals new conclusions and has lower false positive rates, or in which cases there which method is limited to the experimental paradigm used to obtain the data (e.g., how many participants, repetitions, and conditions).

---

## [Author Response]

Essential revisions:The authors appear to get lost in details and some of the key methods are hard to find in the Methods section. This will make the paper hard to follow for those not super familiar with the analysis approach.

We have restructured the manuscript to make it easier to follow. Concretely, we moved the methods parts that are newly introduced here into the main text. We collected all parts that give a general explanation of state of the art representational similarity analysis into a coherent material and methods section (5.1) for readers that are not familiar with the analysis approach yet. And to avoid getting lost in details, we pushed all parts that we did not deem central to our new methods into Appendices. We believe this separation makes the manuscript much more readable.

Additionally, we redistributed the flexible model figure, which one of the reviewers described as the hardest to follow. This is now separated into an early part that explains the method and math and a later section that reports our tests of the method.

Expanding the section on testing for the presence of information would be very useful to broaden the appeal.Expanding the discussion and exposition of the generalizability of the statistical tests (see Reviewer #1).

We discuss this point in a bit more detail in the paper in the new section 3.3. We agree that this is a test that is frequently and sensibly employed and should be mentioned. In the case we are most interested in, this is a relatively low information test, as we should expect most of our models to be distinguishable from pure noise predictions.

We also discuss in this section, what kind of tests our methods cover exactly and which ones we don’t. This should hopefully make things clearer.

The new section 3.3. contains the following text:

“3.3 Supported tests and implications of test results

Our methods enable comparison of a model’s RDM prediction performance (1) against other models, (2) against the noise ceiling, and (3) against chance performance. The first two of these tests are central to the evaluation of models. The test against chance performance is often also reported, but represents a low bar that we should expect most models to pass. In practice, RDM correlations tend to be positive even for very different representations, because physically highly similar stimuli or conditions tend to be similar in all representations. Just like a significant Pearson correlation indicates a dependency, but does not demonstrate that the dependency is linear, a significant RDM prediction result indicates the presence of stimulus information, but does not lend strong support to the particular model. We should resist interpreting significant prediction performance per se as evidence for a particular model (the single-model-significance fallacy; Kriegeskorte and Douglas (2019)). Theoretical progress instead requires that each model be compared to alternative models and to the noise ceiling. An additional point to note is that the interpretation of chance performance, where the RDM comparator equals 0, depends on the chosen RDM comparator, differing, for example, between the Pearson correlation coefficient and the cosine similarity (Diedrichsen et al., 2020).

RDM comparators like the Pearson correlation and the cosine similarity are related to the distance correlation (Székely, Rizzo, and Bakirov, 2007), a general indicator of mutual information. Like a significant distance correlation, a significant RDM correlation mainly demonstrates that there is some mutual information between the brain region in question and the model representation. For a visual representation, for example, all that is required is for the two representations to contain some shared information about the input images. In contrast to the distance correlation (and other non-negative estimates of mutual information), however, negative RDM correlations can occur, indicating simply that pairs of stimuli close in one representation tend to be far in the other and vice versa. For any RDM, there is even a valid perfectly anti-correlated RDM (Pearson r = −1), which can be found by flipping the sign of all dissimilarities and adding a large enough value to make the RDM conform to the triangle inequality (which ensures the existence of an embedding of points that is consistent with the anti-correlated RDM). The existence of valid negative RDM correlations is important to the inferential methods presented here because it is required for our assumption of symmetric (t-)distributions around the true RDM correlation.

Omnibus tests for the presence of information about the experimental conditions in a brain region have been introduced in previous studies (e.g. Allefeld, Görgen, and Haynes, 2016; Kriegeskorte, Goebel, and Bandettini, 2006; Nili, Walther, Alink, and Kriegeskorte, 2020). Whether stimulus information is present in a region is closely related to the question whether the noise ceiling is significantly larger than 0, indicating RDM replicability. Such tests can sensitively detect small amounts of information in the measured activity patterns and can be helpful to assess whether there is any signal for model comparisons. If we are uncertain whether there is a reliable representational geometry to be explained, we need not bother with model comparisons.

The question whether an individual dissimilarity is significantly larger than zero is equivalent to the question whether the distinction between the two conditions can be decoded from the brain-activity. Decoding analyses can be used for this purpose (Hebart, Görgen, and Haynes, 2015; Kriegeskorte and Douglas, 2019; Naselaris, Kay, Nishimoto, and Gallant, 2011; Tong and Pratte, 2012). Such tests require care because the discriminability of two conditions cannot be systematically negative (Allefeld et al., 2016). This is in contrast to comparisons between RDMs, which can be systematically negative (although, as mentioned above, they tend to be positive in practice).”

Reviewer #1 (Recommendations for the authors):1. I really enjoyed reading this manuscript and believe it will make an important contribution to the field. That said, the authors introduce a lot in this work that is only indirectly related to the statistical analysis framework, and as a consequence, the manuscript is currently quite dense and hard to follow. I think that this manuscript would benefit strongly from a much more focused treatment of the key aspects (the introduction of the new method) and a reduction of the emphasis on advanced methods that are not key to this work (such as the use of reweighting and neuronal population sampling approaches, to name only a few).I think the issue is that given the flexibility of analyzing representational geometries with RSA, the authors try to be as general as possible and try to encompass all possible use cases in their writing. In addition, the specific use case for a cross-validated two-factor bootstrap seems to be fitting flexible models, which alone is already quite advanced. I know this is difficult to solve, so I would like to provide one specific recommendation for making the manuscript easier to digest: it would perhaps help to first provide the reader with a quick run-through, without justifying all steps in detail but only summarizing the approach and the basic motivation for it. Then a more thorough treatment, including relevant parts from the methods section that explains the motivation behind the two-factor bootstrap could follow, again followed by extensive validation. This is just one suggestion for improving clarity.

As described in our response to the essential revisions above, we restructured the manuscript to make it less dense. In particular, we tried to move as many of the only indirectly related parts to appendices.

We also followed the reviewers suggestion to move the relevant methods sections into the main text and generated a Materials and methods section that gives a proper exposition to the state of the art RSA methods. As most of the content in that section is not new we decided to place this part into the Materials and methods rather than the Results section. Nonetheless, this part should fulfill the purpose of explaining advanced topics like the new evaluations metrics and flexible models to readers that are not familiar with them.

2. Given the very common use of RSA for testing the presence of effects, rather than model comparison, I think the impact of this work would be strengthened if the authors expanded on their specific use case, even if it is comparably simple (they call this "simple dependence test", which is perhaps a little confusing to the reader).

We hope this point is resolved with our new section 3.3. And the answer we give to the general point above.

3. RSA measures the match of one or more model RDMs with a data RDM. For a test against chance, without very specific biases, a negative representational similarity should not be found empirically for subjects and only for a subset of stimuli. Any such effects should thus only be caused by measurement noise or by stimulus variability. I am wondering to what degree this affects the ability to carry out valid inferences against the null at the population level. See Allefeld et al. (2016) for the treatment of a similar problem with decoding accuracies.

The problem the reviewer highlights here is very important for tests for the presence of any information in the data. We do not cover this kind of test in this manuscript though. We are always testing model performances here and model performances can be reliably negative. We do see that this distinction is fairly subtle and that we did not state this clearly in our previous manuscript. To make this distinction clearer, we added some discussion on which exact tests we cover in this paper in our new section 3.3:

“In contrast to the distance correlation (and other non-negative estimates of mutual information), however, negative RDM correlations can occur, indicating simply that pairs of stimuli close in one representation tend to be far in the other and vice versa. For any RDM, there is even a valid perfectly anti-correlated RDM (Pearson r = −1), which can be found by flipping the sign of all dissimilarities and adding a large enough value to make the RDM conform to the triangle inequality (which ensures the existence of an embedding of points that is consistent with the anti-correlated RDM). The existence of valid negative RDM correlations is important to the inferential methods presented here because it is required for our assumption of symmetric (t-)distributions around the true RDM correlation.”

4. The introduction would benefit from a better motivation of the method. It seems as if the authors discuss previous work on RDMs but then jump to the introduction of the new method. Did no other method exist before? What were the issues with these methods, and what is the gap that needs to be filled? This would help the reader better understand why they should be reading this work.

This connection does indeed seem important. We added a paragraph that explicitly states the limitations of previous RSA inference methods:

“However, existing statistical inference methods for RSA (step 3) have important limitations. Established RSA inference methods (Nili et al., 2014) provide a noise ceiling and enable comparisons of fixed models with generalization to new subjects and conditions. However, they cannot handle flexible models, can be severely suboptimal in terms of statistical power, and have not been thoroughly validated using simulated or real data where ground truth is known. Addressing these shortcomings poses three substantial challenges. (1)

Model-comparative inference with generalization to new conditions is not trivial because new conditions extend an RDM and the evaluation depends on pairwise dissimilarities, thus violating independence assumptions. (2) Standard methods for statistical inference do not handle multiple random factors — subjects and conditions in RSA. (3) Flexible models, that is models that have parameters enabling them to predict different RDMs, are essential for RSA (Diedrichsen, Yokoi, and Arbuckle, 2018; Kriegeskorte and Diedrichsen, 2016). Evaluation of such models requires methods that are unaffected by overfitting to either subjects or conditions to avoid a bias in favor of more flexible models.”

5. While valid, the approach appears to be rather conservative, producing very low false positive rates. Are false negatives not a potentially problematic issue in that respect?

The reviewer here refers to the simple simulations based on matrix normal data, which we now present in Appendix 1—figure 1 for model comparisons. In this case, there were indeed some methods that are overly conservative.

There are three reasons why we believe that this is less problematic than it seems:

1) This impression is due to the simplified simulations. In these simulations we enforced that the two models performed equally well for the exact stimuli in the RDM. This is even more equal than one would expect for a random sample of stimuli for models that perform equally on average. For simulations that include variation due to the stimuli, the methods that look good in our earlier simulations can fail catastrophically, as we demonstrate with a new simulation in which we create such models that perform equally only on average (New third row of Figure 8). Thus the existing, less conservative methods are in fact invalid.

2) The new methods we propose in this manuscript are strictly more progressive than existing methods for the 2-factor bootstrap. We explicitly bound the variance estimates with the uncorrected 2-factor bootstrap. Thus, we produce fewer false negatives than existing valid methods.

3) We show in a diverse set of more realistic simulations, that we can estimate the variance quite accurately. Given an unbiased variance estimate and fairly simple unimodal sampling distributions for the model evaluations, we believe that there is not that much room for improvement left.

Overall, we are sure that we improve over existing methods, also in terms of false negative rates. Nonetheless, it is true that there could be other methods that gain even more power than the methods we propose by raising the false positive rate to the nominal value.

Reviewer #2 (Recommendations for the authors):This paper addresses a major question in computational neuroscience by proposing a novel methodology to test models to explain behavioral/brain data that generalize across conditions and subjects using bootstrapping.The experiments reported validate the claims of the authors. The methodology is applied and analysed in different available datasets.I found particularly helpful and thorough the tests with the simulations. However, I found that the reported analysis is focused mainly on the newly proposed method, and this could bring a wider perspective into the picture.It is with such simulated data that I believe a deeper discussion and possibly adding a comparison to existing methods, such as vanilla RSA and/or linear encoding methods could be reported to support the final discussion on the limitations of such existing methods. This would allow showcasing in which cases this method reveals new conclusions and has lower false positive rates, or in which cases there which method is limited to the experimental paradigm used to obtain the data (e.g., how many participants, repetitions, and conditions).

We believe there are two aspects to this question, which are largely independent:

First, we can compare to other inference methods for RSA to illustrate their limitations. The primary reason to do this is to show that these limited methods indeed fail for the simultaneous generalization to subjects and stimuli that we cover now.

We have added one such simulation result in Appendix 1—figure 1 in the Appendix. There we created two models that are equally close to the data across a large set of stimuli and then subsample these stimuli to create simulated experiments. This leads to vastly inflated false alarm rates for differences between the two models, for all inference methods that do not handle variance due to stimulus selection explicitly. Preliminary simulations based on the deep neural networks also showed that all methods based on the variance across subjects do indeed fail. These simulations drive home the point that earlier methods were insufficient, but we felt they were distracting from our main point that the methods we now recommend work. Also, the severity of these failures depends on how much variability is due to subject choice and stimulus choice respectively and we can make all methods that are based on the variability across subjects fail arbitrarily badly, depending on the simulation parameters, such that the precise size of the error is rather meaningless.

Second, it would be interesting to compare RSA to other methods that are used for comparisons between representational models, like encoding models to estimate which method performs best for model selection. Such comparisons feel out of scope for this manuscript unfortunately. Which model performance metric is best suited for comparing representational models will depend on many technical details of the model evaluations, which we would not want to add to this already dense manuscript. Furthermore, the statistical problems we tackle here apply equally to any other model performance metrics and the question which one works best is largely orthogonal to our questions here.